# Low-rank lottery tickets: finding efficient low-rank neural networks via matrix differential equations

**Steffen Schotthöfer**[*]
Karlsruhe Institute of Technology
76131 Karlsruhe (Germany)
`steffen.schotthoefer@kit.edu`

**Emanuele Zangrando**[*]
Gran Sasso Science Institute
67100 L'Aquila (Italy)
`emanuele.zangrando@gssi.it`

**Jonas Kusch**
University of Innsbruck
6020 Innsbruck (Austria)
`jonas.kusch1@gmail.com`

**Gianluca Ceruti**
EPF Lausanne
1015 Lausanne (Switzerland)
`gianluca.ceruti@epfl.ch`

**Francesco Tudisco**
Gran Sasso Science Institute
67100 L'Aquila (Italy)
`francesco.tudisco@gssi.it`

## Abstract

Neural networks have achieved tremendous success in a large variety of applications. However, their memory footprint and computational demand can render them impractical in application settings with limited hardware or energy resources. In this work, we propose a novel algorithm to find efficient low-rank subnetworks. Remarkably, these subnetworks are determined and adapted already during the training phase and the overall time and memory resources required by both training and evaluating them are significantly reduced. The main idea is to restrict the weight matrices to a low-rank manifold and to update the low-rank factors rather than the full matrix during training. To derive training updates that are restricted to the prescribed manifold, we employ techniques from dynamic model order reduction for matrix differential equations. This allows us to provide approximation, stability, and descent guarantees. Moreover, our method automatically and dynamically adapts the ranks during training to achieve the desired approximation accuracy. The efficiency of the proposed method is demonstrated through a variety of numerical experiments on fully-connected and convolutional networks.

## 1 Introduction

While showing great performance in terms of classification records, most state-of-the-art neural networks require an enormous amount of computation and memory storage both for the training and the evaluation phases [27]. These requirements not only increase infrastructure costs and energy consumption, but also make the deployment of artificial neural networks to infrastructures with limited resources such as mobile phones or smart devices prohibitive. On the other hand, it is well-known that networks' weights contain structures and redundancies that can be exploited for reducing the parameter space dimension without significantly affecting the overall accuracy [3, 9, 17].

Network pruning is a popular line of research that addresses this problem by removing redundant parameters from pre-trained models. Typically, the initial network is large and accurate, and the goal is to produce a smaller network with similar accuracy. Methods within this area include weight sparsification [20, 24, 49] and quantization [10, 60], with different pruning techniques, including search-based heuristics [24], reinforcement learning [2, 23] and genetic algorithms [43]. More recent work has considered pruning during training, by formulating pruning as a data-driven optimization

---

[*]These authors contributed equally to this work.

36th Conference on Neural Information Processing Systems (NeurIPS 2022).

problem [20, 26, 27]. The resulting "dynamical pruning" boils down to a parameter-constrained training phase which, however, has been mostly focused on requiring sparse or binary weights so far.

Rather than enforcing sparsity or binary variables, in this work we constrain the parameter space to the manifold of low-rank matrices. Neural networks' parameter matrices and large data matrices in general are seldom full rank [15, 48, 53, 56]. Constraining these parameters to lie on a manifold defined by low-rank matrices is thus a quite natural approach. By interpreting the training problem as a continuous-time gradient flow, we propose a training algorithm based on the extension of recent Dynamical Low-Rank Approximation (DLRA) algorithms [4, 5, 6]. This approach allows us to use low-rank numerical integrators for matrix Ordinary Differential Equations (ODEs) to obtain modified forward and backward training phases that only use the small-rank factors in the low-rank representation of the parameter matrices and that are stable with respect to small singular values. This is a striking difference with respect to recent alternative "vanilla" low-rank training schemes [31, 57] which simply factorize the weight matrices as the product of two low-rank factors $UV^\top$ and apply a descent algorithm alternatively on the two variables $U$ and $V$.

We perform several experimental evaluations on fully-connected and convolutional networks showing that the resulting dynamical low-rank training paradigm yields low-parametric neural network architectures which compared to their full-rank counterparts are both remarkably less-demanding in terms of memory storage and require much less computational cost to be trained. Moreover, the trained low-rank neural networks achieve comparable accuracy to the original full architecture. This observation is reminiscent of the so-called lottery tickets hypothesis — dense neural networks contain sparse subnetworks that achieve high accuracy [17] — and suggests the presence of *low-rank winning tickets*: highly-performing low-rank subnetworks of dense networks. Remarkably, our dynamical low-rank training strategy seems to be able to find the low-rank winning tickets directly during the training phase independent of initialization.

## 2 Related work on low-rank methods

Low-rank factorization using the SVD and other matrix decomposition techniques have been extensively studied in the scientific computing and machine learning communities. The challenge of compressing and speeding up large-scale neural networks using low-rank methods has sparked wide-spread research interest in recent years and significant effort has been put towards developing low-rank factorization strategies for deep neural networks.

Previous works can roughly be categorized in approaches with fixed low rank and variable low rank during training time. Fixed rank approaches decompose weight matrices using SVD or tensor decompositions of pre-trained networks and fine-tune the factorized network [12, 38, 50, 55], constrain weight matrices to have a fixed low rank during training [30, 31, 57], or create layers as a linear combination of layers of different rank [29]. Hence, these methods introduce the rank of the matrix decomposition as another hyperparameter to be fine-tuned. Rank-adaptive methods mitigate this issue by automatic determination and adaption of the low-rank structure after training. In particular, [33, 34] apply heuristics to determine the rank of the matrix decomposition ahead of time, whereas [59] encourages low-rank weights via a penalized loss that depends on approximated matrix ranks.

Few methods have been proposed recently that adapt the ranks of the weight matrix alongside the main network training phase. In [40], the authors set up the neural network training as a constrained optimization problem with an upper bound on the ranks of the weights, which is solved in an alternating approach resulting in an NP-hard mixed integer program. The authors of [28] formulate a similar constrained optimization problem resulting in a mixed discrete-continuous optimization scheme which jointly addresses the ranks and the elements of the matrices. However, both these approaches require knowledge of the full weight matrix (and of its singular value decomposition) during training and overall are more computational demanding than standard training.

In this work we overcome the above issues and propose a training algorithm with reduced memory and computational requirements. To this end, we reinterpret the optimization problem of a neural network as a gradient flow of the network weight matrices and thus as a matrix ODE. This continuous formulation allows us to use recent advances in DLRA methods for matrix ODEs which aim at evolving the solution of the differential equation on a low-rank manifold. The main idea of DLRA [35], which originates from the Dirac-Frenkel variational principle [14, 18], is to approximate the solution through a low-rank factorization and derive evolution equations for the individual factors.

Thereby, the full solution does not need to be stored and the computational costs can be significantly reduced. To ensure robustness of the method, stable integrators have been proposed in [44] and [6]. Instead of evolving individual low-rank factors in time, these methods evolve products of low-rank factors, which yields remarkable stability and exactness properties [32], both in the matrix and the tensor settings [8, 36, 45, 46]. In this work, we employ the "unconventional" basis update & Galerkin step integrator [6] as well as its rank-adaptive extension [4], see also [7, 37]. The rank-adaptive unconventional integrator chooses the approximation ranks according to the continuous-time training dynamics and allows us to find highly-performing low-rank subnetworks directly during the training phase, while requiring reduced training cost and memory storage.

## 3 Low-rank training via gradient flow

Consider a feed-forward fully-connected neural network $\mathcal{N}(x) = z_M$, with $z_0 = x \in \mathbb{R}^{n_0}$, $z_k = \sigma_k(W_k z_{k-1} + b_k) \in \mathbb{R}^{n_k}$, $k = 1, \ldots, M$ (the convolutional setting is discussed in the supplementary material §6.6). We consider the training of $\mathcal{N}$ based on the optimization of a loss function $\mathcal{L}(W_1, \ldots, W_M; \mathcal{N}(x), y)$ by means of a gradient-based descent algorithm. For example, when using gradient descent, the weights of $\mathcal{N}$ at iteration $t \in \mathbb{N}$ are updated via

$$W_k^{t+1} = W_k^t - \lambda \nabla_{W_k} \mathcal{L}(W_1, \ldots, W_M; \mathcal{N}(x), y) \qquad \forall k = 1, \ldots, M \tag{1}$$

with a learning rate $\lambda$. When the weight matrices $W_k$ are dense, both the forward and gradient evaluations of the network require a large number of full matrix multiplications, with high computational expense and large memory footprint. This renders the training and the use of large-scale neural networks a difficult challenge on limited-resource devices. At the same time, a wealth of evidence shows that dense networks are typically overparameterized and that most of the weights learned this way are unnecessary [15, 48]. In order to reduce the memory and computation costs of training, we propose a method that performs the minimization over the manifold of low-rank matrices.

To this end, we assume that the ideal $W_k$ can be well-approximated by a matrix of rank $r_k \ll n_k, n_{k+1}$ of the form $U_k S_k V_k^\top \in \mathbb{R}^{n_k \times n_{k-1}}$, where $U_k \in \mathbb{R}^{n_k \times r_k}$, $V_k \in \mathbb{R}^{n_{k-1} \times r_k}$ are thin and tall matrices having orthonormal columns spanning optimal subspaces which capture essential properties of parameters, and $S_k \in \mathbb{R}^{r_k \times r_k}$ is a tiny full-rank matrix that allows us to extrapolate the useful information from the learned subspaces $U_k$ and $V_k$.

Traditional descent algorithms such as (1) do not guarantee the preservation of the low-rank structure $U_k S_k V_k^\top$ when updating the weights during training and require knowledge of the whole $W_k$ rather than the factors $U_k, S_k, V_k$. Here we reinterpret the loss minimization as a continuous-time gradient flow and derive a new training method that overcomes all aforementioned limitations.

Minimizing the loss function with respect to $W_k$ is equivalent to evaluating the long time behaviour of the following matrix ODE that allows us to interpret the training phase as a continuous process:

$$\dot{W}_k(t) = -\nabla_{W_k} \mathcal{L}(W_1, \ldots, W_M; \mathcal{N}(x), y), \tag{2}$$

where the "dot" denotes the time-derivative. Let $\mathcal{M}_{r_k}$ denote the manifold of matrices with rank $r_k$ and assume at a certain time $t_0$ the weights are in the manifold, i.e., $W_k(t_0) \in \mathcal{M}_{r_k}$. Using this continuous-time interpretation allows us to derive a strategy to evolve the weights according to the dynamics in (2) so that $W_k(t) \in \mathcal{M}_{r_k}$ for all $t \geq t_0$. To this end, in §3.1 we exploit the fact that $W_k(t)$ admits a time-dependent factorization [13] $W_k(t) = U_k(t) S_k(t) V_k(t)^\top$ to rewrite (2) as a system of matrix ODEs for each of the individual factors $U_k, S_k$ and $V_k$. Then, in §4 we propose an algorithm to efficiently integrate the system of ODEs. We show in §4.1 that such algorithm reduces the loss monotonically and is accurate, in the sense that $U_k S_k V_k^\top \approx W_k$, i.e. the learned low-dimensional subspaces $U_k$ and $V_k$ well-match the behaviour of the full-rank network $W_k$ solution of (2), through the action of the learned $S_k$. Remarkably, using the dynamics of the individual factors will also allow us to adaptively adjust the rank $r_k$ throughout the continuous-time training process.

### 3.1 Coupled dynamics of the low-rank factors via DLRA

We consider the dynamical system of a single weight matrix $W_k$, while the remaining weight matrices are fixed in time and are treated as parameters for the gradient. In the following, we omit writing these parameters down for efficiency of exposition. Assuming $W_k(t) \in \mathcal{M}_{r_k}$, we can formulate (2) as

$$\min \left\{ \|\dot{W}_k(t) + \nabla_{W_k} \mathcal{L}(W_k(t))\|_F \ : \ \dot{W}_k(t) \in \mathcal{T}_{W_k(t)} \mathcal{M}_{r_k} \right\} \tag{3}$$

where $\mathcal{T}_{W_k(t)}\mathcal{M}_{r_k}$ is the tangent space of $\mathcal{M}_{r_k}$ at position $W_k(t)$. In order to solve (3), we further observe that (3) can be equivalently formulated as the following Galerkin condition [35]:

$$\langle \dot{W}_k(t) + \nabla_{W_k}\mathcal{L}(W_k(t)), \delta W_k \rangle = 0 \qquad \forall \delta W_k \in \mathcal{T}_{W_k(t)}\mathcal{M}_{r_k} \ . \tag{4}$$

From $W_k = U_k S_k V_k^\top$, a generic element $\delta W_k$ of the tangent space $\mathcal{T}_{W_k(t)}\mathcal{M}_{r_k}$ can be written as

$$\delta W_k = \delta U_k S_k V_k^\top + U_k \delta S_k V_k^\top + U_k S_k \delta V_k^\top \ ,$$

where $\delta U_k$ and $\delta V_k$ are generic elements of the tangent space of the Stiefel manifold with $r_k$ orthonormal columns at the points $U_k$ and $V_k$, respectively, and $\delta S_k$ is a generic $r_k \times r_k$ matrix, see e.g. [35, §2] for details. Additionally, the Gauge conditions $U_k^\top \delta U_k = 0$ and $V_k^\top \delta V_k = 0$ must be imposed to ensure orthogonality of the basis matrices, and the uniqueness of the representation of the tangent space elements. Similarly, by the chain rule applied several times we have

$$\dot{W}_k = \frac{d}{dt}\{U_k S_k V_k^\top\} = \dot{U}_k S_k V_k^\top + U_k \dot{S}_k V_k^\top + U_k S_k \dot{V}_k^\top \ .$$

Now, the Galerkin condition (4) becomes

$$\langle \dot{U}_k S_k V_k^\top + U_k \dot{S}_k V_k^\top + U_k S_k \dot{V}_k^\top + \nabla_{W_k}\mathcal{L}(W_k(t)), \delta W_k \rangle = 0, \qquad \forall \delta W_k \in \mathcal{T}_{W_k(t)}\mathcal{M}_{r_k} \tag{5}$$

with $U_k^\top \dot{U}_k = 0$ and $V_k^\top \dot{V}_k = 0$. If we choose $\delta W_k = U_k \delta S_k V_k^\top$ in (5), we obtain

$$\langle U_k^\top \dot{U}_k S_k V_k^\top V_k + U_k^\top U_k \dot{S}_k V_k^\top V_k + U_k^\top U_k S_k \dot{V}_k^\top V_k + U_k^\top \nabla_{W_k}\mathcal{L}(W_k(t))V_k, \delta S_k \rangle = 0 \ .$$

Thus, using the Gauge conditions, we obtain $\langle \dot{S}_k + U_k^\top \nabla_{W_k}\mathcal{L}(W_k(t))V_k, \delta S_k \rangle = 0$, which has to hold for a generic $r_k \times r_k$ matrix $\delta S_k$. We obtain this way an evolution equation for the $S_k(t)$ factor. Similarly, specifying (5) for the two choices $\delta W_k = \delta U_k S_k V_k^\top$ and $\delta W_k = U_k S_k \delta V_k^\top$, allows us to obtain the following system of differential equations for the individual factors of $W_k$:

$$\begin{cases} \dot{S}_k = -U_k^\top \nabla_{W_k}\mathcal{L}(W_k(t))V_k \ , \\ \dot{U}_k = -(I - U_k U_k^\top)\nabla_{W_k}\mathcal{L}(W_k(t))V_k S_k^{-1} \ , \\ \dot{V}_k = -(I - V_k V_k^\top)\nabla_{W_k}\mathcal{L}(W_k(t))^\top U_k S_k^{-\top} \ . \end{cases} \tag{6}$$

## 4 KLS-based training algorithm

In order to perform an efficient and robust rank-constrained training step, we numerically integrate the system of ODEs (6). Our approach is based on the "unconventional KLS integrator" [6] and its rank-adaptive version [4]. The pseudocode of the proposed training strategy is presented in Algorithm 1.

The main idea of the KLS algorithm is to alternately represent the product $W_k = U_k S_k V_k^\top$ as $W_k = K_k V_k^\top$ and $W_k = U_k L_k^\top$, consider the corresponding coupled ODEs from (6), and then perform three main steps:

1,2. **K&L-steps** (in parallel). Update the current $K_k$ and $L_k$ by integrating the differential equations

$$\begin{cases} \dot{K}_k(t) = -\nabla_{W_k}\mathcal{L}(K_k(t)V_k^\top)V_k, & K_k(0) = U_k S_k, \\ \dot{L}_k(t) = -\nabla_{W_k}\mathcal{L}(U_k L_k(t)^\top)^\top U_k, & L_k(0) = V_k S_k^\top, \end{cases} \tag{7}$$

from $t = 0$ to $t = \eta$; then form new orthonormal basis matrices $\widetilde{U}_k$ and $\widetilde{V}_k$ spanning the range of the computed $K_k(\eta)$ and $L_k(\eta)$.

3. **S-step**. Update the current $S_k$ by integrating the differential equation

$$\dot{S}_k(t) = -\widetilde{U}_k^\top \nabla_{W_k}\mathcal{L}(\widetilde{U}_k S_k(t)\widetilde{V}_k^\top)\widetilde{V}_k \tag{8}$$

from $t = 0$ to $t = \eta$, with initial value condition $S_k(0) = \widetilde{U}_k^\top U_k S_k V_k^\top \widetilde{V}_k$ .

An important feature of this algorithm is that it can be extended to rank adaptivity in a relatively straightforward manner [4], letting us dynamically evolve the rank of $S_k$ (and thus the rank of $W_k$) during the computation. This is particularly useful, as we may expect the weight matrices to have low ranks but we may not know what the "best" ranks for each layer are. Typically, dynamically adapting

the ranks of a low-rank optimization scheme is a challenging problem as moving from the manifold $\mathcal{M}_{r_k}$ to $\mathcal{M}_{r_k \pm 1}$ introduces singular points [1, 19]. Instead, treating the training problem as a system of matrix differential equations allows us to overcome this issue with a simple trick: at each step of the KLS integrator we double the dimension of the basis matrices $\widetilde{U}_k$ and $\widetilde{V}_k$ computed in the K- and L-steps by computing orthonormal bases spanning $[K_k(\eta) \mid U_k]$ and $[L_k(\eta) \mid V_k]$, respectively, i.e. by augmenting the current basis with the basis computed in the previous time step. Then, after the new $S_k$ matrix is computed via the S-step, a truncation step is performed, removing from the newly computed $S_k$ matrix all the singular values that are under a certain threshold $\vartheta$.

Of course, adding the rank-adaptivity to the integrator comes at a cost. In that case, each step requires to perform an SVD decomposition of twice the size of the current rank of $S_k$ in order to be able to threshold the singular values. Moreover, the dimension of the bases $U_k$ and $V_k$ may grow, which also may require additional computational effort. However, if the ranks remain small throughout the dynamics, this computational overhead is negligible, as we will further discuss in §4.3 and §5.

### 4.1 Error analysis and convergence

In this section we present our main theoretical results, showing that (a) the low-rank matrices $U_k S_k V_k^\top$ formed by the weights' factors computed with Alg. 1 are close to the true solution of (2), and (b) that the loss function decreases during DLRT, provided the singular value threshold $\vartheta$ is not too large, i.e., is bounded by a constant times the square of the time-step size $\eta$ (see Theorem 1). In the version we present here, part of the statements are presented in an informal way for the sake of brevity. We refer to the supplementary material §6.1 for details and for the proofs.

Assume the gradient flow $\mathcal{F}_k(Z) = -\nabla_{W_k} \mathcal{L}(W_1, \ldots, Z, \ldots, W_M, \mathcal{N}(x), y)$ in (2) is locally bounded and locally Lipschitz continuous, with constants $C_1$ and $C_2$, respectively. Then,

**Theorem 1.** *Fixed $x$ and $y$, let $W_k(t)$ be the (full-rank) continuous-time solution of* (2) *and let $U_k, S_k, V_k$ be the factors computed with Algorithm 1 after $t$ steps. Assume that the K,L,S steps* (7) *and* (8) *are integrated exactly from 0 to $\eta$. Assume moreover that, for any $Z \in \mathcal{M}_{r_k}$ sufficiently close to $W_k(t\eta)$, the whole gradient flow $\mathcal{F}_k(Z)$ is "$\varepsilon$-close" to $\mathcal{M}_{r_k}$. Then,*

$$\|U_k S_k V_k^\top - W_k(t\eta)\|_F \leq c_1 \varepsilon + c_2 \eta + c_3 \vartheta/\eta \qquad k = 1, \ldots, M$$

*where the constants $c_1$, $c_2$ and $c_3$ depend only on $C_1$ and $C_2$. In particular, the approximation bound does not depend on the singular values of the exact nor the approximate solution.*

Observe that, while the loss function $\mathcal{L}$ decreases monotonically along any continuous-time solution $W_k(t)$ of (2), it is not obvious that the loss decreases when the integration is done onto the low-rank manifold via Algorithm 1. The next result shows that this is indeed the case, up to terms of the order of the truncation tolerance $\vartheta$. More precisely, we have

**Theorem 2.** *Let $W_k^t = U_k^t S_k^t (V_k^t)^\top$ be the low rank weight matrix computed at step $t$ of Algorithm 1 and let $\mathcal{L}(t) = \mathcal{L}(W_1^t, \ldots, W_M^t, \mathcal{N}(x), y)$. Then, for a small enough time-step $\eta$ we have*

$$\mathcal{L}(t+1) \leq \mathcal{L}(t) - \alpha\eta + \beta\vartheta$$

*where $\alpha$ and $\beta$ are positive constants that do not depend on $t$, $\eta$ and $\vartheta$.*

### 4.2 Efficient implementation of the gradients

All the three K,L,S-steps require the evaluation of the gradient flow of the loss function with respect to the whole matrix $W_k$. Different approaches to efficiently compute this gradient may be used. The strategy we discuss below aims at reducing memory and computational costs by avoiding the computation of the full gradient, working instead with the gradient with respect to the low-rank factors.

To this end, we note that for the K-step it holds $\nabla_{W_k} \mathcal{L}(K_k(t)V_k^\top)V_k = \nabla_{K_k} \mathcal{L}(K_k(t)V_k^\top)$. Hence, the whole gradient can be computed through a forward run of the network with respect to $K_k$

$$z_k = \sigma_k \left( K_k(t)V_k^\top z_{k-1} + b_k \right), \qquad k = 1, \ldots, M \tag{9}$$

and taping the gradient with respect to $K_k$. In this way, the full gradient does not need to be computed and the overall computational costs are comprised of running a forward evaluation while taping gradients with respect to $K_k$, analogously to the traditional back-propagation algorithm. The L- and

**Algorithm 1:** Dynamic Low Rank Training Scheme (DLRT)

---

**Input :** Initial low-rank factors $S_k^0 \sim r_k^0 \times r_k^0$; $U_k^0 \sim n_k \times r_k^0$; $V_k^0 \sim n_{k-1} \times r_k^0$ for $k = 1, \ldots, M$;
       `iter`: maximal number of descent iterations per epoch;
       `adaptive`: Boolean flag that decides whether or not to dynamically update the ranks;
       $\vartheta$: singular value threshold for adaptive procedure.

1 **for** *each* `epoch` **do**
2    **for** $t = 0$ *to* $t = $ `iter` **do**
3      **for** *each layer* $k$ **do**
4        $K_k^t \leftarrow U_k^t S_k^t$                               /* K-step */
5        $K_k^{t+1} \leftarrow$ one-step-integrate$\big\{ \dot{K}(t) = -\nabla_K \mathcal{L}(K(t)(V_k^t)^\top z_{k-1} + b_k^t), K(0) = K_k^t \big\}$
6        $L_k^t \leftarrow V_k^t (S_k^t)^\top$                             /* L-step */
7        $L_k^{t+1} \leftarrow$ one-step-integrate$\big\{ \dot{L}(t) = -\nabla_L \mathcal{L}(U_k^t L(t)^\top z_{k-1} + b_k^t), L(0) = L_k^t \big\}$
8        **if** `adaptive` **then**              /* Basis augmentation step */
9          $K_k^{t+1} \leftarrow [K_k^{t+1} \mid U_k^t]$
10         $L_k^{t+1} \leftarrow [L_k^{t+1} \mid V_k^t]$
11        $U_k^{t+1} \leftarrow$ orthonormal basis for the range of $K_k^{t+1}$       /* S-step */
12        $M_k \leftarrow (U_k^{t+1})^\top U_k^t$
13        $V_k^{t+1} \leftarrow$ orthonormal basis for the range of $L_k^{t+1}$
14        $N_k \leftarrow (V_k^{t+1})^\top V_k^t$
15        $\widetilde{S}_k^t \leftarrow M_k S_k^t N_k^\top$
16        $S_k^{t+1} \leftarrow$ one-step-integrate$\big\{ \dot{S}(t) = -\nabla_S \mathcal{L}\big(U_k^{t+1} S(t)(V_k^{t+1})^\top z_{k-1} + b_k^t\big), S(0) = \widetilde{S}_k^t \big\}$
17        **if** `adaptive` **then**             /* Rank compression step */
18          $P, \Sigma, Q \leftarrow \mathrm{SVD}(S_k^{t+1})$
19          $S_k^{t+1} \leftarrow$ truncate $\Sigma$ using the singular value threshold $\vartheta$
20          $U_k^{t+1} \leftarrow U_k^{t+1} \widetilde{P}$ where $\widetilde{P} = [$first $r_k^{t+1}$ columns of $P]$
21          $V_k^{t+1} \leftarrow V_k^{t+1} \widetilde{Q}$ where $\widetilde{Q} = [$first $r_k^{t+1}$ columns of $Q]$
                                                    /* Bias update step */
22        $b_k^{t+1} \leftarrow$ one-step-integrate$\big\{ \dot{b}(t) = -\nabla_b \mathcal{L}(U_k^{t+1} S_k^{t+1}(V_k^{t+1})^\top z_{k-1} + b(t)), b(0) = b_k^t \big\}$

---

S-steps can be evaluated efficiently in the same manner, by evaluating the network while taping the gradients with respect to $L_k$ and $S_k$, respectively. Hence, instead of a single gradient tape (or chain rule evaluation) of the full weight matrix network, we have three gradient tapes, one for each low rank step, whose combined computational footprint is less than the full matrix tape. We provide detailed formulas for all the three gradient tapes in the supplementary material §6.5.

### 4.3 Implementation details, computational costs and limitations

Each step of Alg. 1 requires the computation of two orthonormal bases for the ranges of $K_k^{t+1}$ and $L_k^{t+1}$. There are of course different techniques to compute such orthonormal matrices. In our implementation we use the QR algorithm, which is known to be one of the most efficient and stable approaches for this purpose. In the adaptive strategy the singular values of $S_k^{t+1}$ are truncated according to a parameter $\vartheta$. To this end, in our implementation, we use the Frobenius norm of $\Sigma$. Precisely, we truncate $\Sigma = \mathrm{diag}(\sigma_i)$ at step 19 of Alg. 1 by selecting the smallest principal $r \times r$ submatrix such that $(\sum_{i \geq r+1} \sigma_i^2)^{1/2} \leq \vartheta$. Finally, one-step-integrate denotes a numerical procedure that integrates the corresponding ODE from time $t = 0$ to $t = \eta$. In practice one can employ different numerical integrators, without affecting the ability of the algorithm to reduce the loss function (see [4, Thm. 5]) while maintaining the low-rank structure. In our implementation we used two methods:

1. Explicit Euler. This method applied to the gradient flow coincides with one step of Stochastic Gradient Descent (SGD), applied to the three factors $K_k, L_k, S_k$ independently.
2. Adam. Here we formally compute the new factors by modifying the explicit Euler step as in the Adam optimization method. Note that Nesterov accelerated SGD is known to coincide with a

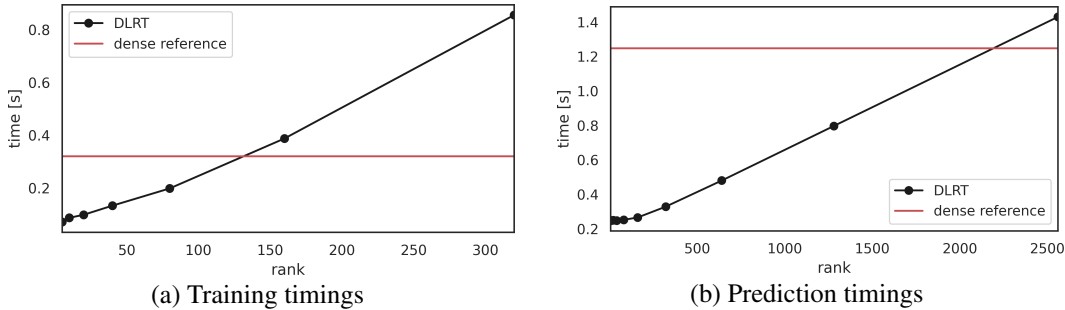

(a) Training timings                  (b) Prediction timings

Figure 1: Comparison of batch execution and training times of 5-layer, 5120-neurons low-rank networks of different ranks and a non-factorized reference network with the same architecture on the MNIST dataset. Training times shown correspond to one epoch for a batch of 256 datapoints. Prediction times refer instead to the whole dataset. All the times are the average over 1000 runs.

particular linear multistep ODE integrator [51]. While Adam does not directly correspond to a numerical integrator to our knowledge, in our tests it resulted in a faster decrease of the loss than both Euler (SGD) and Nesterov accelerated SGD.

For both choices, the target time step $\eta$ corresponds to the value of the learning rate, which we set to 0.2 for Euler. For Adam, we use the default dynamical update, setting 0.001 as starting value.

**Computational cost.** To obtain minimal computational costs and memory requirements for the K-step, the ordering of evaluating $K_k V_k^\top z_{k-1}$ in (9) is important. First, we compute $\widetilde{z} := V_k^\top z_{k-1} \in \mathbb{R}^{r_k}$ which requires $O(r_k n_{k-1})$ operations. Second, we compute $K_k \widetilde{z}$ which requires $O(r_k n_k)$ operations. Adding the bias term and evaluating the activation function requires $O(n_k)$ operations. Hence, combined over all layers we have asymptotic cost of $O\left(\sum_k r_k(n_k + n_{k+1})\right)$. Taping the forward evaluation to compute the gradient with respect to $K_k$ as discussed in §4.2 does not affect the asymptotic costs, i.e. the costs of computing the K-step at layer $k$ assuming a single data point $x$ requires $C_K \lesssim \sum_k r_k(n_k + n_{k+1})$ operations. In a similar manner, we obtain the computational costs of the L- and S-steps, which are again $C_{L,S} \lesssim \sum_k r_k(n_k + n_{k+1})$. Moreover, the QR decompositions used in the K- and L-step require $O\left(\sum_k r_k^2(n_k + n_{k-1})\right)$ operations and computing the SVD in the truncation step has worst-case cost of $O\left(\sum_k r_k^3\right)$. Hence, assuming $r_k \ll n_k, n_{k+1}$, the cost per step of our low-rank method is $C_{\text{DLRA}} \lesssim \sum_k r_k^2(n_k + n_{k-1})$, opposed to the dense network training, which requires $C_{\text{dense}} \lesssim \sum_k n_k n_{k+1}$ operations. In terms of memory cost, note that we only need to store $r_k(r_k + n_k + n_{k+1})$ parameters per layer during the algorithm, corresponding to the matrices $S_k^t, U_k^t, V_k^t$. Moreover, at the end of the training we can further compress memory by storing the product of the trained weight factors $U_k S_k$, rather than the individual matrices.

**Limitations.** A requirement for DLRT's efficiency is that $r_k \ll n_k, n_{k+1}$. When the truncation threshold $\vartheta$ is too small, Alg. 1 does not provide advantages with respect to standard training. This is also shown by Fig. 1. Moreover, in the terminology of [54], DLRT is designed to reduce training costs corresponding to model parameters and to the optimizer. To additionally decrease activation costs, DLRT can be combined with micro-batching or checkpointing approaches. Finally, the choice of $\vartheta$ introduces one additional hyperparameter which at the moment requires external knowledge for tuning. However, our experiments in §5 show that relatively large values of $\vartheta$ yield competing performance as compared to a number of baselines, including standard training.

## 5 Numerical Results

We illustrate the performance of DLRT Algorithm 1 on several test cases. The code is implemented in both Tensorflow (https://github.com/CSMMLab/DLRT-Net) and PyTorch (https://github.com/COMPiLELab/DLRT-Net). The networks are trained on an AMD Ryzen 9 3950X CPU and a Nvidia RTX 3090 GPU. Timings are measured on pure CPU execution.

### 5.1 Performance analysis on MNIST dataset

We partition MNIST dataset [11] in randomly sampled train-validation-test sets of size $50K$-$10K$-$10K$. Images are pixelwise normalized; no further data augmentation or regularization has been used.

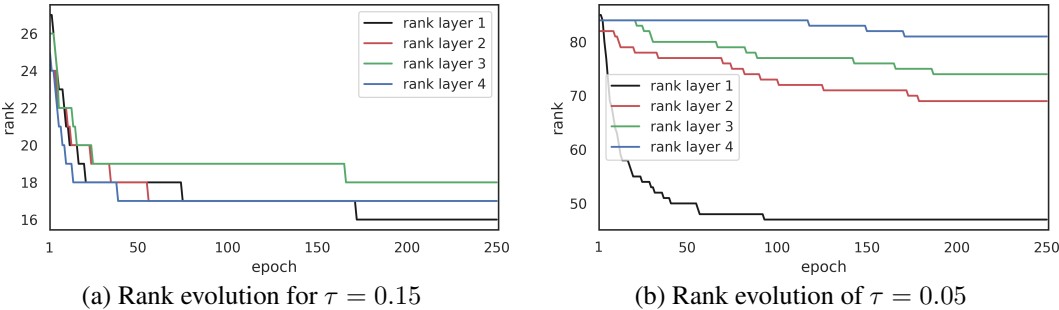

(a) Rank evolution for $\tau = 0.15$       (b) Rank evolution of $\tau = 0.05$

Figure 2: Rank evolution (layers 1-4) of 5-layer [500,500,500,500,10] fully-connected net on MNIST.

**Fixed-rank fully-connected feed-forward network timings.** First, we compare the training time of the adaptive DLRT Alg. 1 on a 5-layer fully-connected $[5120, 5120, 5120, 5120, 10]$ network fixing the ranks of layers 1-4, i.e. choosing a specific starting rank $r_k^0$ for the input weight factors and truncating $\Sigma$ at line 19 of Alg. 1 to the principal $r_k^0 \times r_k^0$ submatrix, rather than via a threshold. Next, we measure the average prediction time on the whole MNIST dataset over 1000 runs. Fig. 1(a) and 1(b) show that both timings scale linearly with the rank of the factorizations, and that for sufficiently small ranks DLRT is faster than the full-rank baseline both in terms of training and of prediction.

**Rank evolution and performance of DLRT for different singular value thresholds.** Next, we demonstrate the capabilities of DLRT to determine the rank of the network's weight matrices automatically during the network training using Algorithm 1. The Adam optimizer with default learning rate is used for the gradient update. We train fully connected 5-layer networks, of which the first 4 are replaced by low-rank layers in the subsequent tests. The activation function is chosen to be ReLU for the hidden layers, and softmax for the output layer. The training loss is sparse categorical cross entropy and we additionally measure the model's accuracy. We use batch size 256 and train for 250 epochs. We choose $\vartheta = \tau \|\Sigma\|$, thus we truncate the singular values of the current $S_k^t$ by a fraction $\tau$ of the total Frobenius norm. The smaller $\tau$, the more singular values are kept.

Figures 2 (a) and (b) show the evolution of the rank adaptive layers of a 5-layer 500-neuron network in a long time case study for $\tau = 0.05$ and $\tau = 0.15$. We can see that within the first epoch the initial full matrix ranks are reduced significantly, to 27 for $\tau = 0.15$, and to $\sim 85$ for $\tau = 0.05$ respectively. Within the first 50 epochs, the layer ranks are already close to their final ranks. This indicates that the rank adaptive algorithm is only needed for the first few training epochs, and can then be replaced by the computationally cheaper fixed-low-rank training (by setting the Boolean variable `adaptive` to `False` in Algorithm 1). Figure 3 compares the mean test accuracy of 5-layer networks with 500 and 784 neurons with different levels of low-rank compression, over five independent runs with randomly sampled train-test-val sets. The networks can be compressed via dynamical low-rank training by more than 95%, while only losing little more than 1% test accuracy compared to the dense reference network marked in red. Remark that restricting the space of possible networks to a given rank regularizes the problem, since such a restriction can be understood as adding a PCR regularization term to the loss function. This can be seen from the tendency of not overfitting and reaching improved test accuracies compared to the corresponding dense network for moderate compression ratios. Also note that adaptive-low rank training eliminates the need for hyperparameter grid search in terms of layer-weights, due to automatic rank adaptation. The rank dynamics for all configurations can be seen in the supplementary material §6.3. Finally, in the supplementary material §6.4 we compare the use of DLRT with the vanilla approach which simply thresholds the singular values of the full-rank network. Our results show that advantageous low-rank winning tickets exist, but are not easy to find. In fact, the vanilla low-rank subnetworks perform very poorly. From this point of view, our approach can be seen as an efficient dynamical pruning technique, able to determine high-performing low-rank subnetworks in a given dense network. Remarkably, our numerical experiments suggest that low-rank winning tickets can be trained from the start and do not to heavily depend on the initial weight guess.

**Convolutional layers: LeNet5.** Here we compare the proposed dynamical low-rank training scheme on LeNet5 [39] on MNIST, against the full-rank reference and several baselines. SVD prune [61] and LRNN [28] are the closest approaches to our DLRT: they dynamically train low-rank layers by adding a rank-penalty to the loss function, and by complementing the standard training step via an

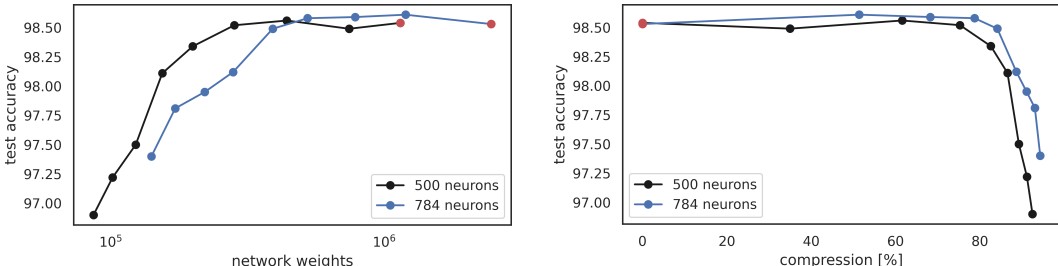

Figure 3: Mean test accuracy over parameters' number and compression rate for 5 runs with randomly sampled train-test-val sets on 5-layer fully-connected nets. Red dots denote the full-rank baseline.

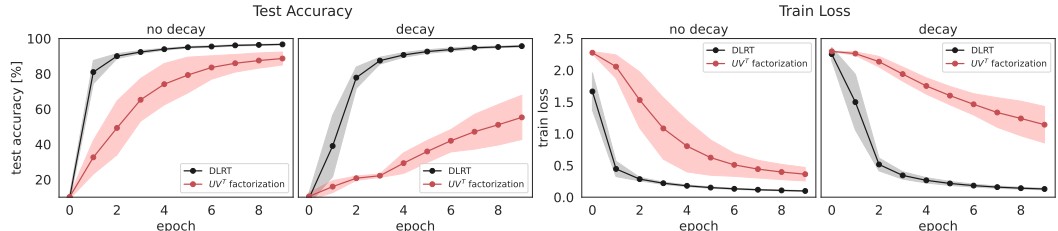

Figure 4: Mean learning curves with standard deviation of Lenet5 on MNIST over 10 runs of DLRT compared to a vanilla layer factorization $W_k = U_k V_k^\top$. Both methods are implemented with fixed learning rate of $0.01$, and batch size of $128$. The weight matrices are either completely randomly initialized ("no decay") or are initialized with a random choice forced to have an exponential decay on the singular values ("decay").

SVD projection step in the latter and a pruning step in the former. While computing low-rank factors for each layer, thus reducing memory storage of the network, this training approach is more expensive than training the full network. GAL [42], SSL [62], and NISP [58] are pruning methods which aim at learning optimal sparse weights (rather than low-rank) by adding sparsity-promoting regularization terms to the training loss. As for LRNN, these methods do not reduce the computational cost of the training phase (as indicated with the $< 0\%$ in Table 1). Analogously to [28], our adaptive low-rank training technique is applied to the convolutional layers by flattening the tensor representing the convolutional kernel into a matrix. Details are provided in the supplementary material §6.6. All the models are trained for $120$ epochs using SGD with a fixed learning rate of $0.2$. Results in Table 1 show that the DLRT algorithm is able to find low-rank subnetworks with up to $96.4\%$ less parameters than the full reference, while keeping the test accuracy above $95\%$. Compared to the baseline methods, we achieve better compression rates but observe lower accuracy. However, unlike the baseline references, DLRT automatically prunes the singular values during training, without requirement to solve any additional optimization problem, thus significantly improving time and memory efficiency of both forward and backward phases, with respect to the full reference.

**Robustness with respect to small singular values and comparison with vanilla low-rank parametrization.** A direct way to perform training enforcing a fixed rank for the weight matrices is to parameterize each weight as $W_k = U_k V_k^\top$ and alternating training with respect to $U_k$ and to $V_k$. This is the strategy employed for example in [31, 57]. This vanilla low-rank parametrization approach has a number of disadvantages with respect to DLRT, on top of the obvious non-adaptive choice of the rank. First, DLRT guarantees approximation and descent via Theorems 1 and 2. Second, we observe that the vanilla factorization gives rise to an ill-conditioned optimization method when small singular values occur. This problem is peculiar to the low-rank manifold itself [16, 35], whose local curvature is proportional to the inverse of the smallest singular value of the weight matrices. In contrast, the numerical integration strategy at the basis of DLRT is designed to take advantage of the structure of the manifold and is robust with respect to small singular values [32]. This can be seen from the bound of Theorem 1, where the constants are independent of the singular values of the weight matrices, and is illustrated by Figure 4, where DLRT shows a much faster convergence rate with respect to vanilla SGD performed on each factor of the parametrization $U_k V_k^\top$, when applied to train LeNet5 on MNIST. Both methods are implemented with the same fixed learning rate.

Table 1: Results of the training of LeNet5 on MNIST dataset. Effective parameters represent the number of parameters we have to save for evaluating the network and those we need in order to train via the DLRT Alg.1. The compression ratio (c.r.) is the percentage of parameter reduction with respect to the full model ($< 0\%$ indicates that the ratio is negative). "ft" indicates that the model has been fine-tuned. "LeNet5" denotes the standard LeNet5 architecture trained with SGD.

| | method | NN metrics | | Evaluation | | Train | |
| | | test acc. | ranks | params | c.r. | params | c.r. |
|---|---|---|---|---|---|---|---|
| | LeNet5 | **99.2%** | $[20, 50, 500, 10]$ | 430500 | 0% | 430500 | 0% |
| DLRT | $\tau = 0.11$ | 98.0% | $[15, 46, 13, 10]$ | 47975 | 88.86% | 50585 | 88.25% |
| | $\tau = 0.15$ | 97.8% | $[13, 31, 9, 10]$ | 34435 | 92.0% | 35746 | 91.7% |
| | $\tau = 0.2$ | 97.2% | $[10, 20, 7, 10]$ | 25650 | 94.04% | 26299 | 93.89% |
| | $\tau = 0.3$ | 95.3% | $[6, 9, 4, 10]$ | 15520 | **96.4%** | 15753 | **96.34%** |
| | SSL [62] (ft) | 99.18% | | 110000 | 74.4% | | $< 0\%$ |
| | NISP [58] (ft) | 99.0% | | 100000 | 76.5% | | $< 0\%$ |
| | GAL [42] | 98.97% | | 30000 | 93.0% | | $< 0\%$ |
| | LRNN [28] | 98.67% | $[3, 3, 9, 9]$ | 18075 | 95.8% | | $< 0\%$ |
| | SVD prune [61] | 94.0% | $[2, 5, 89, 10]$ | 123646 | 71.2% | | $< 0\%$ |

Table 2: Results on ImageNet1k (left) and Cifar10 (right). The compression ratio is the percentage of parameter reduction with respect to the full model. DLRT is used with $\tau = 0.1$. The number of parameters of the full models are: 33.6M (VGG16); 23.6M (AlexNet); 29.6M (ResNet-50). We report difference in test accuracy (top-5 test accuracy for ImageNet1k) with respect to the full baselines.

| | | ImageNet1k | | |
| | | test acc.[%] | compression rate | |
| | method | (to baseline) | eval[%] | train[%] |
|---|---|---|---|---|
| ResNet-50 | DLRT | $-0.56$ | 54.1 | 14.2 |
| | PP-2[52] | $-0.8$ | 52.2 | $< 0$ |
| | PP-1[52] | $-0.2$ | 44.2 | $< 0$ |
| | CP[25] | $-1.4$ | 50.0 | $< 0$ |
| | SFP[22] | $-0.2$ | 41.8 | $< 0$ |
| | ThiNet[47] | $-1.5$ | 36.9 | $< 0$ |
| VGG16 | DLRT | $-2.19$ | 86 | 78.4 |
| | PP-1[52] | $-0.19$ | 80.2 | $< 0$ |
| | CP[25] | $-1.80$ | 80.0 | $< 0$ |
| | ThiNet[47] | $-0.47$ | 69.04 | $< 0$ |
| | RNP(3X)[41] | $-2.43$ | 66.67 | $< 0$ |

| | | Cifar10 | | |
| | | test acc.[%] | compression rate | |
| | method | (to baseline) | eval[%] | train[%] |
|---|---|---|---|---|
| VGG16 | DLRT | $-1.89$ | 56 | 77.5 |
| | GAL[42] | $-1.87$ | 77 | $< 0$ |
| | LRNN[28] | $-1.9$ | 60 | $< 0$ |
| AlexNet | DLRT | $-1.79$ | 86.3 | 84.2 |
| | NISP[58] | $-1.06$ | $-$ | $< 0$ |

## 5.2 Results on ImageNet1K and Cifar10 with ResNet-50, AlexNet, and VGG16

Finally, we assess the capability of compressing different architectures on large scale training sets. We train a full-rank baseline model and compare it to DLRT using the same starting weights on an Nvidia A-100 GPU. The used optimizer is SGD with momentum factor $0.1$ and no data-augmentation techniques are used. We compare the results on ResNet-50, VGG16, and AlexNet models, on the Cifar10 and ImageNet1k datasets, and with respect to a number of low-parametric alternative baselines methods. For DLRT, the last layers of the networks have been adapted to match the corresponding classification tasks. Detailed results are reported in Table 2, where we show the test accuracy (reported as the difference with respect to the full baseline) as well as compression ratios. With Cifar10, we archive a train compression of $77.5\%$ with an accuracy loss of just $1.89\%$ for VGG16 and $84.2\%$ train compression at $1.79\%$ accuracy loss for AlexNet. In the ImageNet1k benchmark, we achieve a train compression rate of $14.2\%$, with an test accuracy loss of $0.5\%$ in top-5 accuracy on ResNet-50 and $78.4\%$ train compression with $2.19$ top-5 accuracy loss on VGG16.

**Acknowledgements.** The work of S. Schotthöfer was funded by the Priority Programme SPP2298 "Theoretical Foundations of Deep Learning" by the Deutsche Forschungsgemeinschaft (DFG). The work of J. Kusch was funded by the Deutsche Forschungsgemeinschaft (DFG) – 491976834. The work of G. Ceruti was supported by the SNSF research project "Fast algorithms from low-rank updates", grant number 200020-178806. The work of F. Tudisco and E. Zangrando was funded by the MUR-PNRR project "Low-parametric machine learning". Special thanks to Prof. Martin Frank for the PhD mentorship of Steffen Schottöfer.

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
