

(a) Discrete time weight update

(b) Continuous time weight update

(c) Galerkin condition

Figure 5: Panels (a)-(b): Graphical re-interpretation of the weight update step as a time-continuous process. Panel (c): Orthogonal projection onto the tangent space of the low-rank manifold $\mathcal{M}_r$. The dashed line depicts the projection resulting in $\dot{W}_k(t)$, which is the tangent element minimizing the distance between $\nabla_{W_k}\mathcal{L}(W_k(t))$ and the tangent space $\mathcal{T}_{W_k(t)}\mathcal{M}_r$ at the approximation $W_k(t)$.

## 6  Appendix

### 6.1  Proofs of the main results

We provide here a proof of Theorems 1 and 2. The proof is based on a number of classical results as well as recent advances in DLRA theory, including [4, 6, 32, 35, 44].

Recall that, for a fixed layer $k$, we reinterpret the training phase as a continuous-time evolution of the weights on the manifold of low-rank matrices, as illustrated in Fig. 5(a-b). This boils down to solving the manifold-constrained matrix differential equation

$$\min\left\{\|\dot{W}_k(t) - \mathcal{F}_k(W_k(t))\| \,:\, \dot{W}_k(t) \in \mathcal{T}_{W_k(t)}\mathcal{M}_{r_k}\right\},\qquad(10)$$

where $\|\cdot\|$ is the Frobenius norm and $\mathcal{F}_k$ denotes the gradient flow of the loss with respect to the $k$-th matrix variable, namely

$$\mathcal{F}_k(Z) = -\nabla_{W_k}\mathcal{L}(W_1,\ldots,Z,\ldots,W_M;\mathcal{N}(x),y)\,.$$

For the sake of simplicity and for a cleaner notation, as all the results we will present hold for a generic $k$, we drop the subscript $k$ from now on. In particular, we assume $W$ is the weight matrix of a generic hidden layer with $n$ input and $m$ output neurons.

In order for our derivation to hold, we require the following two properties:

P1.  The gradient flow $\mathcal{F}$ is locally bounded and locally Lipschitz continuous, with constants $C_1$ and $C_2$, respectively. Namely, we assume there exist $C_1, C_2 > 0$ (independent of $k$) such that

$$\|\mathcal{F}(Z)\| \leq C_1 \qquad \|\mathcal{F}(Z) - \mathcal{F}(\tilde{Z})\| \leq C_2\|Z - \tilde{Z}\|$$

for all $Z, \tilde{Z} \in \mathbb{R}^{m\times n}$.

P2.  The whole gradient flow is "not too far" from the rank-$r$ manifold $\mathcal{M}_r$. Precisely, we assume that for any $Z \in \mathcal{M}_r$ arbitrary close to $W(t)$, the whole gradient flow $\mathcal{F}(Z)$ near $t$ is such that

$$\|(I - \mathrm{P}(Z))\mathcal{F}(Z)\| \leq \varepsilon\,,$$

where $\mathrm{P}(Z)$ denotes the orthogonal projection onto $\mathcal{T}_Z\mathcal{M}_r$.

Note that both assumptions are valid for low-rank neural network training. In particular, Lipschitz continuity and boundedness of the gradient are standard assumptions in optimization and are satisfied by the gradient of commonly used neural networks' losses. Moreover, assuming the gradient flow to be close to the low-rank manifold is an often encountered empirical observation in neural networks [15, 48, 53].

In order to derive the proof of Theorems 1 and 2 we first present a number of relevant background lemmas. The first lemma shows that the subspace generated by the K-step in Algorithm 1 after the QR-decomposition is $\mathcal{O}(\eta(\eta + \varepsilon))$ close to the range of the exact solution, where $\eta$ is the time-step of the integrator and $\varepsilon$ is the eigenvalue truncation tolerance.

**Lemma 1** ([6, Lemma 2]). *Let $W^1$ be the solution at time $t = \eta$ of the full problem (2) with initial condition $W^0$. Let $U^1$ be the matrix obtained with the K-step of the fixed-rank Algorithm 1, after one step. Under the assumptions P1 and P2 above, we have*

$$\|U^1 U^{1,\top} W^1 - W^1\| \leq \theta$$

*where*

$$\theta = C_1 C_2 (4e^{C_2\eta} + 9)\eta^2 + (3e^{C_2\eta} + 4)\varepsilon\eta.$$

*Proof.* The local error analysis of [32] shows that there exists $L^1$ such that

$$\|U^1 L^{1,\top} - W^1\| \leq \theta.$$

It follows that,

$$\begin{aligned}
\|U^1 L^{1,\top} - W^1\|^2 &= \|U^1 L^{1,\top} - U^1 U^{1,\top} W^1 + U^1 U^{1,\top} W^1 - W^1\|^2 \\
&= \|U^1 U^{1,\top}(U^1 L^{1,\top} - W^1) + (I - U^1 U^{1,\top})(-W^1)\|^2 \\
&= \|U^1 U^{1,\top}(U^1 L^{1,\top} - W^1)\|^2 + \|(I - U^1 U^{1,\top})W^1\|^2.
\end{aligned}$$

Therefore,

$$\|U^1 U^{1,\top}(U^1 L^{1,\top} - W^1)\|^2 + \|(I - U^1 U^{1,\top})W^1\|^2 \leq \theta^2.$$

Hence, since both terms must be bounded by $\theta^2$ individually, we obtain the stated result. $\qquad\square$

In the next lemma we show that also the space generated by the $L$ step is close by the exact solution. Namely, combined with the previous result, we have

**Lemma 2** ([6, Lemma 3]). *Let $W^1$, $U^1$ be defined as above. Let $V^1$ be the matrix obtained from the L-step of the fixed-rank Algorithm 1, after one step. The following estimate holds:*

$$\|U^1 U^{1,\top} W^1 V^1 V^{1,\top} - W^1\| \leq 2\theta.$$

*Proof.* The L-step is obtained as the K-step applied to the transposed function $\mathcal{G}(Y) = \mathcal{F}(Y^\top)^\top$. Due to the invariance of the Frobenius norm under transposition, property P1 holds. Similarly, property P2 continues to be satisfied because

$$\|(I - \mathrm{P}(Y))\mathcal{G}(Y)\| = \|(I - \mathrm{P}(Y^\top))\mathcal{F}(Y^\top)\| \leq \varepsilon,$$

where the equality $\mathrm{P}(Y)Z^\top = \left[\mathrm{P}(Y^\top)Z\right]^\top$ has been used [35, §4]. It follows from Lemma 1 that

$$\begin{aligned}
\|U^1 U^{1,\top} W^1 - W^1\| &\leq \theta, \\
\|V^1 V^{1,\top} W^{1,\top} - W^{1,\top}\| &\leq \theta.
\end{aligned} \tag{11}$$

This implies that

$$\begin{aligned}
\|U^1 U^{1,\top} W^1 V^1 V^{1,\top} - W^1\| &\leq \|U^1 U^{1,\top} W^1 V^1 V^{1,\top} - W^1 V^1 V^{1,\top} + W^1 V^1 V^{1,\top} - W^1\| \\
&\leq \|U^1 U^{1,\top} W^1 V^1 V^{1,\top} - W^1 V^1 V^{1,\top}\| + \|W^1 V^1 V^{1,\top} - W^1\| \\
&\leq \|(U^1 U^{1,\top} W^1 - W^1)V^1 V^{1,\top}\| + \|V^1 V^{1,\top} W^{1,\top} - W^{1,\top}\| \\
&\leq \|U^1 U^{1,\top} W^1 - W^1\| \cdot \|V^1 V^{1,\top}\|_2 + \|V^1 V^{1,\top} W^{1,\top} - W^{1,\top}\|.
\end{aligned}$$

Because $\|V^1 V^{1,\top}\|_2 = 1$, the stated result follows from (11). $\qquad\square$

With the previous lemmas, we are in the position to derive the local error bound for the fixed-rank KLS integrator of Section 4.

**Lemma 3** (Local Error, [6, Lemma 4]). *Let $W^1, U^1, V^1$ be defined as above and let $S^1$ be the matrix obtained with the S-step of Algorithm 1 after one step. The following local error bound holds:*

$$\|U^1 S^1 V^{1,\top} - W^1\| \leq \eta(\hat{c}_1 \varepsilon + \hat{c}_2 \eta),$$

*where the constants $\hat{c}_i$ are independent of the singular values of $W^1$ and $S^1$.*

*Proof.* From Lemma 2 and the equality $Y^1 = U^1 S^1 V^{1,\top}$, we have that

$$\|Y^1 - W^1\| \leq \|Y^1 - U^1 U^{1,\top} W^1 V^1 V^{1,\top}\| + \|U^1 U^{1,\top} W^1 V^1 V^{1,\top} - W^1\|$$
$$\leq \|U^1(S^1 - U^{1,\top} W^1 V^1)V^{1,\top}\| + 2\theta$$
$$\leq \|S^1 - U^{1,\top} W^1 V^1\| + 2\theta.$$

The local error's analysis is reduced to bound the term $\|S^1 - U^{1,\top} W^1 V^1\|$. For $0 \leq t \leq \eta$, we thus introduce the following auxiliary quantity:

$$\widetilde{S}(t) := U^{1,\top} W(t) V^1.$$

We observe that the term $W(t)$ can be re-written as

$$W(t) = U^1 U^{1,\top} W(t) V^1 V^{1,\top} + \left(W(t) - U^1 U^{1,\top} W(t) V^1 V^{1,\top}\right) = U^1 \widetilde{S}(t) V^{1,\top} + \mathcal{R}(t),$$

where $\mathcal{R}(t)$ denotes the term in big brackets. For $0 \leq t \leq \eta$, it follows from Lemma 2 and the bound $C_1$ of the function $\mathcal{F}$ that

$$\|W(t) - W(\eta)\| \leq \int_0^\eta \|\dot{W}(s)\| \, ds = \int_0^\eta \|\mathcal{F}(W(s))\| \, ds \leq C_1 \eta.$$

Therefore, the term $\mathcal{R}(t)$ is bounded by

$$\|\mathcal{R}(t)\| \leq \|\mathcal{R}(t) - \mathcal{R}(\eta)\| + \|\mathcal{R}(\eta)\| \leq 2C_1 \eta + 2\theta.$$

We re-cast the function $\mathcal{F}(W(t))$ as

$$\mathcal{F}(W(t)) = \mathcal{F}(U^1 \widetilde{S}(t) V^{1,\top} + \mathcal{R}(t))$$
$$= \mathcal{F}(U^1 \widetilde{S}(t) V^{1,\top}) + \mathcal{D}(t)$$

where the defect $\mathcal{D}(t)$ is given by

$$\mathcal{D}(t) := \mathcal{F}(U^1 \widetilde{S}(t) V^{1,\top} + \mathcal{R}(t)) - \mathcal{F}(U^1 \widetilde{S}(t) V^{1,\top}).$$

Via the Lipschitz constant $C_2$ of the function $\mathcal{F}$, the defect is bounded by

$$\|\mathcal{D}(t)\| \leq C_2 \|\mathcal{R}(t)\| \leq 2C_2(C_1 \eta + \theta).$$

Now, we compare the two differential equations

$$\dot{\widetilde{S}}(t) = U^{1,\top} \mathcal{F}(U^1 \widetilde{S}(t) V^{1,\top}) V^1 + U^{1,\top} \mathcal{D}(t) V^1, \qquad \widetilde{S}(0) = U^{1,\top} W^0 V^1,$$
$$\dot{S}(t) = U^{1,\top} \mathcal{F}(U^1 S(t) V^{1,\top}) V^1, \qquad\qquad\qquad S(0) = U^{1,\top} W^0 V^1.$$

The solution $S^1$ obtained in the second differential equation is the same as given by the S-step of the KLS integrator of Section 4. By construction, the solution obtained in the first differential equation at time $t = \eta$ is $\widetilde{S}(\eta) = U^{1,\top} W^1 V^1$. With the Gronwall inequality we obtain

$$\|S^1 - U^{1,\top} W^1 V^1\| \leq \int_0^\eta e^{C_2(\eta-s)} \|\mathcal{D}(s)\| \, ds \leq e^{L\eta} 2C_2(C_1 \eta + \theta)\eta.$$

The result yields the statement of the theorem using the definition of $\theta$. $\qquad\square$

We are now in the position to conclude the proof of Theorem 1.

*Proof of Theorem 1.* In Lemma 3, the local error for the fixed-rank integrator of §4 has been provided. The local error in time of the rank-adaptive version is directly obtained via a triangle inequality:

$$\|U^1 S^1 V^{1,\top} - W(\eta)\| \leq \hat{c}_1 \varepsilon \eta + \hat{c}_2 \eta^2 + \vartheta\,,$$

where $\vartheta$ is the tolerance parameter chosen for the truncation procedure. Here, we abuse the notation and we maintain the same nomenclature $U^1, S^1$, and $V^1$ also for the novel low-rank approximation obtained via the truncation procedure.

Thus, we conclude the proof using the Lipschitz continuity of the function $\mathcal{F}$. We move from the local error in time to the global error in time by a standard argument of Lady Windermere's fan [21, Section II.3]. Therefore, the error after $t$ steps of the rank-adaptive Algorithm 1 is given by

$$\|U^t S^t V^{t,\top} - W(t\eta)\| \leq c_1 \varepsilon + c_2 \eta + c_3 \vartheta/\eta\,.$$

$\qquad\square$

To conclude with, we prove that after one step the proposed rank-adaptive DLRT algorithm decreases along the low-rank approximations. We remind that only property P1 needs to be assumed here.

*Proof of Theorem 2.* Let $\widehat{Y}(t) = U^1 S(t) V^{1,\top}$. Here, $S(t)$ denotes the solution for $t \in [0, \eta]$ of the S-step of the rank-adaptive integrator . It follows that

$$
\begin{aligned}
\frac{d}{dt} \mathcal{L}(\widehat{Y}(t)) &= \langle \nabla \mathcal{L}(\widehat{Y}(t)), \dot{\widehat{Y}}(t) \rangle \\
&= \langle \nabla \mathcal{L}(\widehat{Y}(t)), U^1 \dot{S}(t) V^{1,\top} \rangle \\
&= \langle U^{1,\top} \nabla \mathcal{L}(\widehat{Y}(t)) V^1, \dot{S}(t) \rangle \\
&= \langle U^{1,\top} \nabla \mathcal{L}(\widehat{Y}(t)) V^1, -U^{1,\top} \nabla \mathcal{L}(\widehat{Y}(t)) V^1 \rangle = -\| U^{1,\top} \nabla \mathcal{L}(\widehat{Y}(t)) V^1 \|^2 .
\end{aligned}
$$

The last identities follow by definition of the S-step. For $t \in [0, \eta]$ we have

$$
\frac{d}{dt} \mathcal{L}(\widehat{Y}(t)) \leq -\alpha^2 \tag{12}
$$

where $\alpha = \min_{0 \leq \tau \leq 1} \| U^{1,\top} \nabla \mathcal{L}(\widehat{Y}(\tau\eta)) V^1 \|$. Integrating (12) from $t = 0$ until $t = \eta$, we obtain

$$
\mathcal{L}(\widehat{Y}^1) \leq \mathcal{L}(\widehat{Y}^0) - \alpha^2 \eta.
$$

Because the subspace $U^1$ and $V^1$ contain by construction the range and co-range of the initial value, we have that $\widehat{Y}^0 = U^0 S^0 V^{0,\top}$ [4, Lemma 1]. The truncation is such that $\| Y^1 - \widehat{Y}^1 \| \leq \vartheta$. Therefore,

$$
\mathcal{L}(Y^1) \leq \mathcal{L}(\widehat{Y}^1) + \beta\vartheta
$$

where $\beta = \max_{0 \leq \tau \leq 1} \| \nabla \mathcal{L}(\tau Y^1 + (1 - \tau)\widehat{Y}^1) \|$. Hence, the stated result is obtained. $\qquad\square$

## 6.2 Detailed timing measurements

Table 3 displays the average batch training times of a 5-layer, 5120-neuron dense network on the MNIST dataset, with a batch size of 500 samples. We average the timings over 200 batches and additionally display the standard deviation of the timings corresponding to the layer ranks. The batch timing measures the full K,L and S steps, including back-propagation and gradient updates, as well as the loss and metric evaluations.

Table 3: Average batch training times for fixed low-rank training of a 5-layer fully-connected network with layer widths $[5120, 5120, 5120, 5120, 10]$. Different low-rank factorizations are compared

| ranks | mean time [s] | std. deviation [s] |
|---|---|---|
| full-rank | 0.320 | $\pm 0.005227$ |
| $[320, 320, 320, 320, 320]$ | 0.855 | $\pm 0.006547$ |
| $[160, 160, 160, 160, 10]$ | 0.387 | $\pm 0.005657$ |
| $[80, 80, 80, 80, 10]$ | 0.198 | $\pm 0.004816$ |
| $[40, 40, 40, 40, 10]$ | 0.133 | $\pm 0.005984$ |
| $[20, 20, 20, 20, 10]$ | 0.098 | $\pm 0.005650$ |
| $[10, 10, 10, 10, 10]$ | 0.087 | $\pm 0.005734$ |
| $[5, 5, 5, 5, 10]$ | 0.071 | $\pm 0.005369$ |

Table 4 shows the average test time of a 5-layer, 5120-neuron dense network, for different low-rank factorizations and the full rank reference network. The timings are averaged over 1000 evaluations of the $60K$ sample MNIST training data set. We measure the $K$ step forward evaluation of the low-rank networks as well as the loss and prediction accuracy evaluations.

## 6.3 Detailed training performance of adaptive low-rank networks

Tables 5 and 6 display a detailed overview of the adaptive low-rank results of §5.1. The displayed ranks are the ranks of the converged algorithm. The rank evolution of the 5-Layer, 500-Neuron test case can be seen in Fig. 6. The Evaluation parameter count corresponds to the parameters of the

Table 4: Average dataset prediction times for fixed low-rank training of a 5-layer fully-connected network with layer widths $[5120, 5120, 5120, 5120, 10]$. Different low-rank factorizations are compared.

| ranks | mean time [s] | std. deviation [s] |
|---|---|---|
| full-rank | 1.2476 | $\pm 0.0471$ |
| $[2560, 2560, 2560, 2560, 10]$ | 1.4297 | $\pm 0.0400$ |
| $[1280, 1280, 1280, 1280, 10]$ | 0.7966 | $\pm 0.0438$ |
| $[640, 640, 640, 640, 10]$ | 0.4802 | $\pm 0.0436$ |
| $[320, 320, 320, 320, 10]$ | 0.3286 | $\pm 0.0442$ |
| $[160, 160, 160, 160, 10]$ | 0.2659 | $\pm 0.0380$ |
| $[80, 80, 80, 80, 10]$ | 0.2522 | $\pm 0.0346$ |
| $[40, 40, 40, 40, 10]$ | 0.2480 | $\pm 0.0354$ |
| $[20, 20, 20, 20, 10]$ | 0.2501 | $\pm 0.0274$ |
| $[10, 10, 10, 10, 10]$ | 0.2487 | $\pm 0.0276$ |
| $[5, 5, 5, 5, 10]$ | 0.2472 | $\pm 0.0322$ |

$K$ step of the dynamical low-rank algorithm, since all other matrices are no longer needed in the evaluation phase. The training parameter count is evaluated as the number of parameters of the $S$ step of the adaptive dynamical low rank training, with maximal basis expansion by $2r$, where $r$ is the current rank of the network. We use the converged ranks of the adaptive low-rank training to compute the training parameters. Note that during the very first training epochs, the parameter count is typically higher until the rank reduction has reached a sufficiently low level.

Table 5: Dynamical low rank training for 5-layer $500$-neurons network. c.r. denotes the compression rate relative to the full rank dense network.

| | NN metrics | | Evaluation | | Train | |
|---|---|---|---|---|---|---|
| $\tau$ | test acc. | ranks | params | c.r. | params | c.r. |
| full-rank | $98.54 \pm 0.03\%$ | $[500, 500, 500, 500, 10]$ | 1147000 | $0\%$ | 1147000 | $0\%$ |
| 0.03 | $98.49 \pm 0.02\%$ | $[176, 170, 171, 174, 10]$ | 745984 | $34.97\%$ | 1964540 | $-71.27\%$ |
| 0.05 | $98.56 \pm 0.02\%$ | $[81, 104, 111, 117, 10]$ | 441004 | $61.56\%$ | 1050556 | $8.40\%$ |
| 0.07 | $98.52 \pm 0.08\%$ | $[52, 67, 73, 72, 10]$ | 283768 | $75.26\%$ | 633360 | $44.78\%$ |
| 0.09 | $98.34 \pm 0.14\%$ | $[35, 53, 51, 46, 10]$ | 199940 | $82.57\%$ | 429884 | $62.52\%$ |
| 0.11 | $98.11 \pm 0.46\%$ | $[27, 40, 37, 38, 10]$ | 154668 | $86.52\%$ | 324904 | $71.67\%$ |
| 0.13 | $97.50 \pm 0.23\%$ | $[20, 31, 32, 30, 10]$ | 123680 | $89.22\%$ | 255500 | $77.72\%$ |
| 0.15 | $97.22 \pm 0.29\%$ | $[17, 25, 26, 24, 10]$ | 101828 | $91.13\%$ | 207320 | $81.92\%$ |
| 0.17 | $96.90 \pm 0.45\%$ | $[13, 21, 24, 20, 10]$ | 86692 | $92.45\%$ | 174728 | $84.76\%$ |

Table 6: Dynamical low rank training for 5-layer $784$-neurons network. c.r. denotes the compression rate relative to the full rank dense network.

| | NN metrics | | Evaluation | | Train | |
|---|---|---|---|---|---|---|
| $\tau$ | test acc. | ranks | params | c.r. | params | c.r. |
| full-rank | $98.53 \pm 0.04\%$ | $[784, 784, 784, 784, 10]$ | 2466464 | $0\%$ | 2466464 | $0\%$ |
| 0.03 | $98.61 \pm 0.07\%$ | $[190, 190, 190, 190, 10]$ | 1199520 | $51.37\%$ | 2968800 | $-20.36\%$ |
| 0.05 | $98.59 \pm 0.06\%$ | $[124, 120, 125, 126, 10]$ | 784000 | $68.22\%$ | 1805268 | $26.80\%$ |
| 0.07 | $98.58 \pm 0.03\%$ | $[76, 86, 85, 83, 10]$ | 525280 | $78.71\%$ | 1151864 | $53.29\%$ |
| 0.09 | $98.49 \pm 0.05\%$ | $[56, 67, 63, 59, 10]$ | 392000 | $84.41\%$ | 836460 | $66.08\%$ |
| 0.11 | $98.12 \pm 0.21\%$ | $[35, 49, 47, 43, 10]$ | 280672 | $88.63\%$ | 584240 | $76.31\%$ |
| 0.13 | $97.95 \pm 0.23\%$ | $[29, 35, 38, 34, 10]$ | 221088 | $91.04\%$ | 453000 | $81.63\%$ |
| 0.15 | $97.81 \pm 0.17\%$ | $[22, 29, 27, 27, 10]$ | 172480 | $93.01\%$ | 348252 | $85.88\%$ |
| 0.17 | $97.40 \pm 0.25\%$ | $[17, 23, 22, 23, 10]$ | 141120 | $94.28\%$ | 281724 | $88.57\%$ |

### 6.3.1 Lenet5 experiment

In Table 7 we report the results of five independent runs of the dynamic low-rank training scheme on Lenet5; we refer to §5.1 for further details. For each column of the table, we report the mean value together with its relative standard deviations. No seed has been applied for splitting the dataset and generating the initial weights configuration.

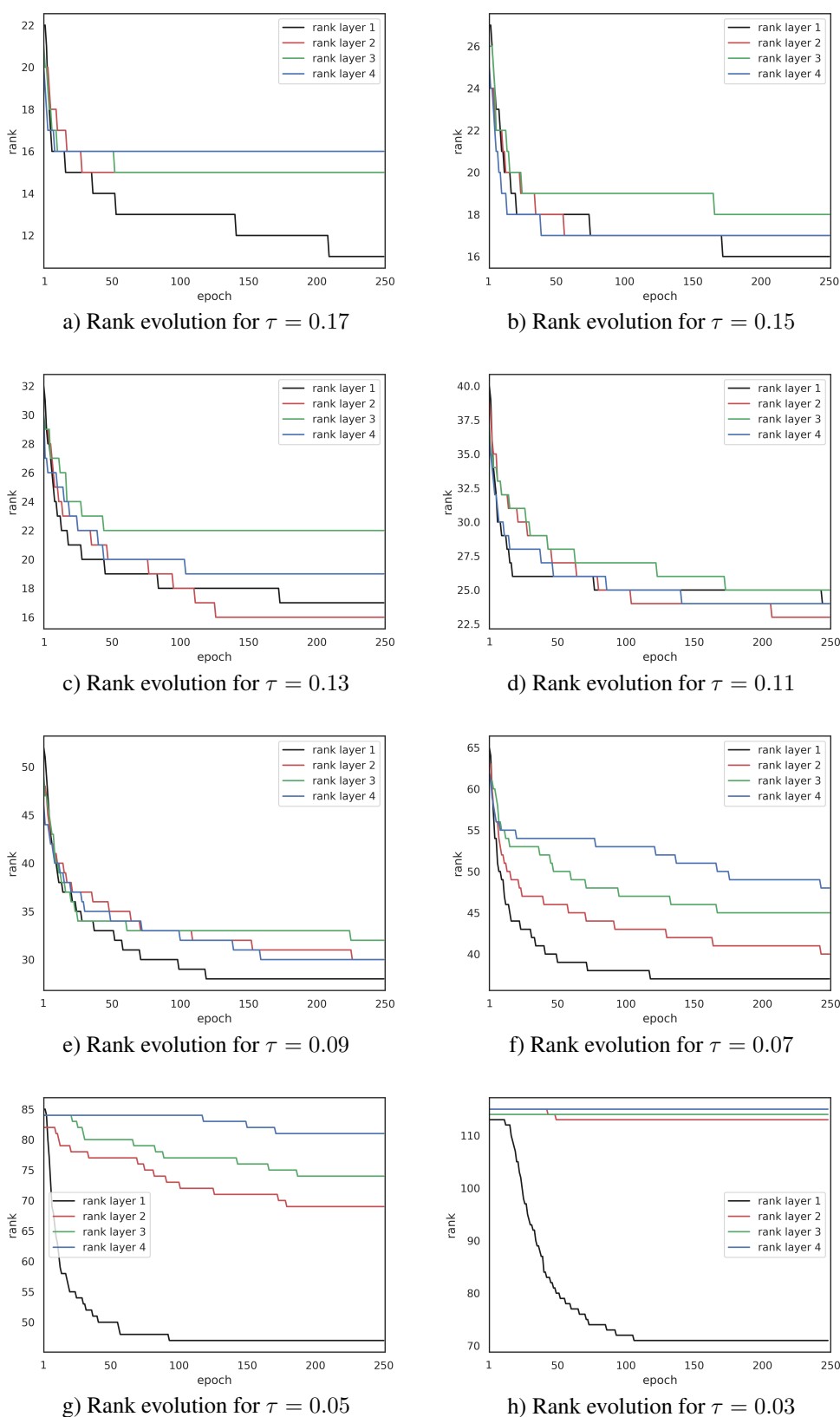

Figure 6: Rank evolution of the dynamic adaptive low-rank training algorithm for the 5-layer, 500-neuron dense architecture.

Table 7: Mean results and standard relative deviations of the dynamic low-rank training algorithm over five independent runs on Lenet5. Adaptive learning rate of $0.05$ with $0.96-$exponentially decaying tax.

| $\tau$ | NN metrics | | Evaluation | | Train | |
| | test acc. | ranks | params | c.r. | params | c.r. |
|---|---|---|---|---|---|---|
| 0.11 | $95.420 \pm 1.865\%$ | $[15, 46, 13, 10]$ | 47975 | 88.9% | 50585 | 88.2% |
| 0.15 | $95.527 \pm 1.297\%$ | $[13, 31, 9, 10]$ | 34435 | 92.0% | 35746 | 91.69% |
| 0.2 | $95.009 \pm 1.465\%$ | $[10, 20, 7, 10]$ | 25650 | 94.04% | 26299 | 93.89% |
| 0.3 | $92.434 \pm 1.757\%$ | $[6, 9, 4, 10]$ | 15520 | 96.39% | 15753 | 96.34% |

## 6.4 Detailed training performance of low-rank pruning

The proposed low-rank training algorithm does not need to be applied to train a network from random initial weight guesses. When an already trained network is available, the proposed method can be employed as a memory-efficient pruning strategy. A straightforward approach to reduce a trained fully-connected network to a rank $r$ network is to compute an SVD for all weight matrices and to truncate those decompositions at rank $r$. However, while this choice is optimal to present weight matrices, it might significantly reduce the accuracy of the network. Hence, retraining the determined low-rank subnetwork is commonly necessary to obtain desirable accuracy properties. Three key aspects are important to obtain an efficient pruning method for low-rank methods:

1. Retraining preserves the low-rank structure of the subnetwork.

2. Retraining does not exhibit the memory footprint of the fully connected network.

3. Retraining finds the optimal network among possible low rank networks.

Let us note that the attractor of the proposed dynamical low-rank evolution equations fulfills these three requirements. Recall that for the evolution equations we have (3):

$$\min \left\{ \|\dot{W}_k(t) + \nabla_{W_k} \mathcal{L}(W_k(t))\|_F \ : \ \dot{W}_k(t) \in \mathcal{T}_{W_k(t)} \mathcal{M}_{r_k} \right\}. \tag{13}$$

The condition $\dot{W}_k(t) \in \mathcal{T}_{W_k(t)} \mathcal{M}_{r_k}$ ensures that the weight matrices remain of low-rank. Moreover, as previously discussed, the training method only requires memory capacities to store low-rank factors. At the attractor, i.e., when $\dot{W}_k = 0$, the last condition ensures that the attractor minimizes $\|\nabla_{W_k} \mathcal{L}(W_k(t))\|_F$. That is, the attractor is the optimal low-rank subnetwork in the sense that it picks the network with minimal gradient. To underline the effectiveness of our low-rank method as a pruning technique, we take the fully connected network from Table 6. To demonstrate the poor validation accuracy when simply doing an SVD on the full $784$ by $784$ weight matrices and truncating at a given smaller rank, we perform this experiment for ranks $r \in \{10, 20, 30, 40, 50, 60, 70, 80, 90, 100\}$. It turns out that though reducing memory requirements, this strategy leads to unsatisfactory accuracy of about $10\%$, see the first column of Table 8. Then, we use the proposed low-rank training methods with fixed rank $r$ to retrain the network. As starting points, we use the low-rank networks which have been determined by the truncated SVD. Retraining then reaches desired accuracies that are comparable to the previously determined low-rank networks in Table 6.

## 6.5 Detailed derivation of the gradient

In this section, we derive the computation of the gradients in the K, L and S steps in detail. For this, let us start with the full gradient, i.e., the gradient of the loss with respect to the weight matrix $W_k$. We have

$$\partial_{W_{jk}^\ell} \mathcal{L} = \sum_{i_M=1}^{n_M} \partial_{z_{i_M}^M} \mathcal{L} \partial_{W_{jk}^\ell} z_{i_M}^M = \sum_{i_M=1}^{n_M} \partial_{z_{i_M}^M} \mathcal{L} \partial_{W_{jk}^\ell} \sigma_M \left( \sum_{i_{M-1}} W_{i_M i_{M-1}} z_{i_{M-1}}^{M-1} + b_{i_M}^M \right)$$

$$= \sum_{i_M=1}^{n_M} \partial_{z_{i_M}^M} \mathcal{L} \sigma_M' \left( \sum_{i_{M-1}} W_{i_M i_{M-1}} z_{i_{M-1}}^{M-1} + b_{i_M}^M \right) \partial_{W_{jk}^\ell} \left( \sum_{i_{M-1}} W_{i_M i_{M-1}} z_{i_{M-1}}^{M-1} \right). \tag{14}$$

Table 8: Pruning methods with 784 Neurons per layer

| test accuracy | | | Evaluation | |
| SVD | low-rank training | ranks | params | c.r. |
| --- | --- | --- | --- | --- |
| 98.63% | 98.63% | $[784, 784, 784, 784, 10]$ | 2466464 | 0% |
| 9.91% | 98.16% | $[100, 100, 100, 100, 10]$ | 635040 | 74.25% |
| 9.67% | 98.44% | $[90, 90, 90, 90, 10]$ | 572320 | 76.80% |
| 9.15% | 98.47% | $[80, 80, 80, 80, 10]$ | 509600 | 79.34% |
| 9.83% | 98.58% | $[70, 70, 70, 70, 10]$ | 446880 | 81.88% |
| 9.67% | 98.41% | $[60, 60, 60, 60, 10]$ | 384160 | 84.42% |
| 9.83% | 98.39% | $[50, 50, 50, 50, 10]$ | 321440 | 86.97% |
| 10.64% | 98.24% | $[40, 40, 40, 40, 10]$ | 258720 | 89.51% |
| 10.3% | 98.24% | $[30, 30, 30, 30, 10]$ | 196000 | 92.05% |
| 9.15% | 97.47% | $[20, 20, 20, 20, 10]$ | 133280 | 94.60% |
| 10.9% | 95.36% | $[10, 10, 10, 10, 10]$ | 70560 | 97.14% |

For a general $\alpha$, let us define

$$
\sigma'_{\alpha, i_\alpha} := \sigma'_\alpha \left( \sum_{i_{\alpha-1}} W^\alpha_{i_\alpha i_{\alpha-1}} z^{\alpha-1}_{i_{\alpha-1}} + b^\alpha_{i_\alpha} \right)
\tag{15}
$$

and note that for $\alpha \neq \ell$

$$
\partial_{W^\ell_{jk}} \left( \sum_{i_{\alpha-1}} W^\alpha_{i_\alpha i_{\alpha-1}} z^{\alpha-1}_{i_{\alpha-1}} \right) = \sum_{i_{\alpha-1}} W^\alpha_{i_\alpha i_{\alpha-1}} \partial_{W^\ell_{jk}} z^{\alpha-1}_{i_{\alpha-1}},
\tag{16}
$$

whereas for $\alpha = \ell$ we have

$$
\partial_{W^\ell_{jk}} \left( \sum_{i_{\alpha-1}=1}^{n_{\alpha-1}} W^\alpha_{i_\alpha i_{\alpha-1}} z^{\alpha-1}_{i_{\alpha-1}} \right) = \sum_{i_{\alpha-1}} \delta_{j i_\alpha} \delta_{k i_{\alpha-1}} z^{\alpha-1}_{i_{\alpha-1}}.
\tag{17}
$$

Therefore, recursively plugging (15), (16) and (17) into (14) yields

$$
\begin{aligned}
\partial_{W^\ell_{jk}} \mathcal{L} &= \sum_{i_M=1}^{n_M} \partial_{z^M_{i_M}} \mathcal{L} \sigma'_{M, i_M} \sum_{i_{M-1}} W^\alpha_{i_M i_{M-1}} \partial_{W^\ell_{jk}} z^{M-1}_{i_{M-1}} \\
&= \sum_{i_M=1}^{n_M} \partial_{z^M_{i_M}} \mathcal{L} \sigma'_{M, i_M} \sum_{i_{M-1}} W^\alpha_{i_M i_{M-1}} \sigma'_{M-1, i_{M-1}} \sum_{i_{M-2}} W^\alpha_{i_{M-1} i_{M-2}} \partial_{W^\ell_{jk}} z^{M-2}_{i_{M-2}} = \cdots \\
&= \sum_{i_\ell, \cdots, i_M} \partial_{z^M_{i_M}} \mathcal{L} \prod_{\alpha=\ell+1}^{M} \sigma'_{\alpha, i_\alpha} W^\alpha_{i_\alpha i_{\alpha-1}} \sigma'_{\ell, i_\ell} \partial_{W^\ell_{jk}} \left( \sum_{i_{\ell-1}=1}^{n_{\ell-1}} W^\ell_{i_\ell i_{\ell-1}} z^{\ell-1}_{i_{\ell-1}} \right) \\
&= \sum_{i_\ell, \cdots, i_M} \partial_{z^M_{i_M}} \mathcal{L} \prod_{\alpha=\ell+1}^{M} \sigma'_{\alpha, i_\alpha} W^\alpha_{i_\alpha i_{\alpha-1}} \sigma'_{\ell, i_\ell} \sum_{i_{\ell-1}} \delta_{j i_\ell} \delta_{k i_{\ell-1}} z^{\ell-1}_{i_{\ell-1}} \\
&= \sum_{i_{\ell+1}, \cdots, i_M} \partial_{z^M_{i_M}} \mathcal{L} \prod_{\alpha=\ell+1}^{M} \sigma'_{\alpha, i_\alpha} W^\alpha_{i_\alpha i_{\alpha-1}} \sigma'_{\ell, j} \delta_{j i_\ell} z^{\ell-1}_k
\end{aligned}
\tag{18}
$$

Written in matrix notation and making use of the Hadamard product defined as $y \circ A \circ x = (y_i A_{ij} x_j)_{ij}$, for $A \in \mathbb{R}^{m \times n}$, $x \in \mathbb{R}^n$ and $y \in \mathbb{R}^m$, we have:

$$
\partial_{W^\ell} \mathcal{L} = \partial_{z^M} \mathcal{L}^\top \left( \sigma'_\ell \circ \prod_{\alpha=\ell+1}^{M} W^\top_\alpha \circ \sigma'_\alpha \right)^\top z^\top_{\ell-1}
$$

Now, let us move to deriving the K, L and S-steps for the dynamical low-rank training. For the K-step, we represent the weight matrix $W_\ell$ as $W^\ell_{i_\ell i_{\ell-1}} = \sum_m K^\ell_{i_\ell m} V^\ell_{i_{\ell-1} m}$. Hence, reusing the intermediate result (18) yields

$$\partial_{K^\ell_{jk}} \mathcal{L} = \sum_{i_\ell, \cdots, i_M} \partial_{z_{i_M}^M} \mathcal{L} \prod_{\alpha=\ell+1}^{M} \sigma'_{\alpha, i_\alpha} W^\alpha_{i_\alpha i_{\alpha-1}} \sigma'_{\ell, i_\ell} \partial_{K^\ell_{jk}} \left( \sum_{i_{\ell-1}=1}^{n_{\ell-1}} \sum_m K^\ell_{i_\ell m} V^\ell_{i_{\ell-1} m} z^{\ell-1}_{i_{\ell-1}} \right)$$

$$= \sum_{i_\ell, \cdots, i_M} \partial_{z_{i_M}^M} \mathcal{L} \prod_{\alpha=\ell+1}^{M} \sigma'_{\alpha, i_\alpha} W^\alpha_{i_\alpha i_{\alpha-1}} \sigma'_{\ell, i_\ell} \sum_{i_{\ell-1}=1}^{n_{\ell-1}} \sum_m \delta_{j i_\ell} \delta_{km} V^\ell_{i_{\ell-1} m} z^{\ell-1}_{i_{\ell-1}}$$

$$= \sum_{i_{\ell+1}, \cdots, i_M} \partial_{z_{i_M}^M} \mathcal{L} \prod_{\alpha=\ell+1}^{M} \sigma'_{\alpha, i_\alpha} W^\alpha_{i_\alpha i_{\alpha-1}} \sigma'_{\ell, i_\ell} \sum_{i_{\ell-1}=1}^{n_{\ell-1}} \delta_{j i_\ell} V^\ell_{i_{\ell-1} k} z^{\ell-1}_{i_{\ell-1}}$$

In matrix notation we obtain

$$\partial_{K_\ell} \mathcal{L} = \partial_{z^M} \mathcal{L}^\top \left( \sigma'_\ell \circ \prod_{\alpha=\ell+1}^{M} W^\top_\alpha \circ \sigma'_\alpha \right)^\top \left( V^\top_\ell z_{\ell-1} \right)^\top = \partial_{W_\ell} \mathcal{L} V_\ell,$$

which is exactly the right-hand side of the K-step. Hence, the K-step can be computed by a forward evaluation of $\mathcal{L}$ and recording the gradient tape with respect to $K^\ell$. Similarly, for the L-step, we represent $W_\ell$ as $W^\ell_{i_\ell i_{\ell-1}} = \sum_m U^\ell_{i_\ell m} L^\ell_{i_{\ell-1} m}$. Hence,

$$\partial_{L^\ell_{jk}} \mathcal{L} = \sum_{i_\ell, \cdots, i_M} \partial_{z_{i_M}^M} \mathcal{L} \prod_{\alpha=\ell+1}^{M} \sigma'_{\alpha, i_\alpha} W^\alpha_{i_\alpha i_{\alpha-1}} \sigma'_{\ell, i_\ell} \partial_{L^\ell_{jk}} \left( \sum_{i_{\ell-1}=1}^{n_{\ell-1}} \sum_m U^\ell_{i_\ell m} L^\ell_{i_{\ell-1} m} \right)$$

$$= \sum_{i_\ell, \cdots, i_M} \partial_{z_{i_M}^M} \mathcal{L} \prod_{\alpha=\ell+1}^{M} \sigma'_{\alpha, i_\alpha} W^\alpha_{i_\alpha i_{\alpha-1}} \sigma'_{\ell, i_\ell} \sum_{i_{\ell-1}=1}^{n_{\ell-1}} \sum_m U^\ell_{i_\ell m} \delta_{j i_{\ell-1}} \delta_{km} z^{\ell-1}_{i_{\ell-1}}$$

$$= \sum_{i_\ell, \cdots, i_M} \partial_{z_{i_M}^M} \mathcal{L} \prod_{\alpha=\ell+1}^{M} \sigma'_{\alpha, i_\alpha} W^\alpha_{i_\alpha i_{\alpha-1}} \sigma'_{\ell, i_\ell} U^\ell_{i_\ell m} z^{\ell-1}_j.$$

In matrix notation, we obtain

$$\partial_{L_\ell} \mathcal{L} = \left( U^\top_\ell \partial_{z^M} \mathcal{L}^\top \left( \sigma'_\ell \circ \prod_{\alpha=\ell+1}^{M} W^\top_\alpha \circ \sigma'_\alpha \right)^\top z^\top_{\ell-1} \right)^\top = (\partial_{W_\ell} \mathcal{L})^\top U_\ell.$$

Lastly, for the S-step we write $W^\ell_{i_\ell i_{\ell-1}} = \sum_{n,m} U^\ell_{i_\ell m} S_{mn} V^\ell_{i_{\ell-1} n}$. Then,

$$\partial_{S^\ell_{jk}} \mathcal{L} = \sum_{i_\ell, \cdots, i_M} \partial_{z_{i_M}^M} \mathcal{L} \prod_{\alpha=\ell+1}^{M} \sigma'_{\alpha, i_\alpha} W^\alpha_{i_\alpha i_{\alpha-1}} \sigma'_{\ell, i_\ell} \partial_{S^\ell_{jk}} \left( \sum_{n,m} U^\ell_{i_\ell m} S_{mn} V^\ell_{i_{\ell-1} n} \right)$$

$$= \sum_{i_\ell, \cdots, i_M} \partial_{z_{i_M}^M} \mathcal{L} \prod_{\alpha=\ell+1}^{M} \sigma'_{\alpha, i_\alpha} W^\alpha_{i_\alpha i_{\alpha-1}} \sigma'_{\ell, i_\ell} \sum_{i_{\ell-1}=1}^{n_{\ell-1}} \sum_m U^\ell_{i_\ell m} \delta_{jm} \delta_{kn} V^\ell_{i_{\ell-1} n} z^{\ell-1}_{i_{\ell-1}}$$

$$= \sum_{i_\ell, \cdots, i_M} \partial_{z_{i_M}^M} \mathcal{L} \prod_{\alpha=\ell+1}^{M} \sigma'_{\alpha, i_\alpha} W^\alpha_{i_\alpha i_{\alpha-1}} \sigma'_{\ell, i_\ell} U^\ell_{i_\ell j} V^\ell_{i_{\ell-1} k} z^{\ell-1}_{i_{\ell-1}}.$$

In matrix notation, we have

$$\partial_{S_\ell} \mathcal{L} = U^\top_\ell \partial_{z^M} \mathcal{L}^\top \left( \sigma'_\ell \circ \prod_{\alpha=\ell+1}^{M} W^\top_\alpha \circ \sigma'_\alpha \right)^\top \left( V^\top_\ell z_{\ell-1} \right)^\top = U^\top_\ell \partial_{W_\ell} \mathcal{L} V_\ell.$$

## 6.6 Low-rank matrix representation and implementation of convolutional layers

A generalized convolutional filter is a four-mode tensor $W \in \mathbb{R}^{F \times C \times J \times K}$ consisting of $F$ filters of shape $C \times J \times K$, which is applied to a batch of $N$ input $C-$ channels image signals $Z$ of spatial dimensions $U \times V$ as the linear mapping,

$$(Z * W)(n, f, u, v) = \sum_{j=1}^{J} \sum_{k=1}^{K} \sum_{c=1}^{C} W(f, c, j, k) Z(n, c, u - j, v - k). \tag{19}$$

In order to train the convolutional filter on the low-rank matrix manifold, we reshape the tensor $W$ into a rectangular matrix $W^{\text{resh}} \in \mathbb{R}^{F \times CJK}$. This reshaping is also considered in e.g. [28]. An option is, to see the convolution as the contraction between an three-mode tensor $Z^{\text{unfolded}}$ of patches and the reshaped kernel matrix $W^{\text{resh}}$ using Pytorch's fold-unfold function. We can construct the unfold by stacking the vectorized version of sliding patterns of the kernel on the original input, obtaining in this way a tensor $Z^{\text{unfolded}} \in \mathbb{R}^{N \times CJK \times L}$, where $L$ denotes the dimension of flatten version of the output of the 2-D convolution. Thus, equation 19 can be rewritten as a tensor mode product:

$$(Z * W)(n, f, u, v) = \sum_{j=1}^{J} \sum_{k=1}^{K} \sum_{c=1}^{C} W^{\text{resh}}(f, (c, j, k)) Z^{\text{unfolded}}(n, (c, j, k), (u, v))$$

$$= \sum_{p=}^{r} U(f, p) \sum_{q=1}^{r} S(p, q) \sum_{j=1}^{J} \sum_{k=1}^{K} \sum_{c=1}^{C} V((c, j, k), q) Z^{\text{unfolded}}(n, (c, j, k), (u, v)) \tag{20}$$

As it is shown in (20), we can a decompose the starting weight $W^{\text{resh}} = USV^{\top}$ and then do all the training procedure as a function of the factors $(U, S, V)$, without ever reconstructing the kernel. Then we can apply the considerations of fully connected layers.