# OpenReview forum: "Low-rank lottery tickets: finding efficient low-rank neural networks via matrix differential equations"
_NeurIPS.cc/2022/Conference — NeurIPS 2022 Accept_

### Official Review · Reviewer_kDdv · 2022-07-10

**Rating:** 4
**Confidence:** 5
**Soundness:** 2 fair
**Presentation:** 2 fair
**Contribution:** 2 fair

**Summary:**

This paper presents a Dynamic Low Rank Training Scheme (DLRT) algorithm, which aims at attaining low-rank networks during training directly for faster computation efficiency and lower memory footprint. Low-rank training is formulated via gradient workflow, then via incorporating the dynamics of the model weights, the proposed DLRT algorithms achieve adjusting the layer ranks dynamically (via explicit optimization). The complexity and efficiency of the DLRT algorithms are discussed. Empirical simulation is also provided to justify the effectiveness and efficiency of the proposed DLRT algorithm.

**Questions:**

Please refer to the "Weaknesses" part for answering my major questions. In summary:

1. Is DLRT designed for efficient training or efficient inference, or both?
2. How does DLRT compare against DeepCompression and EarlyBird Ticket?
3. Why can't DLRT outperform LRNN? If not, why should people use DLRT in practice?
4. DLRT scale to larger networks, e.g., ResNet-50, EfficientNet, ViT, BERT on larger tasks, e.g., ImageNet, WikiText, GLUE benchmark?

**Limitations:**

As discussed thoroughly in the section "Strengths And Weaknesses" and "Questions", the paper is not a theory paper. But in practice, the proposed DLRT method has a few flaws, e.g., computation efficiency, and training-time memory cost is not clear. The empirical results of this paper are weak, the experiments are only conducted on one toy dataset (even though, DLRT is off compared to at least one of the baseline methods).

**Strengths And Weaknesses:**

Strengths:
Overall the proposed idea makes sense. The DLRT algorithm is a viable algorithm for attaining low-rank neural networks (presumably for faster inference after training). The simulated experiments support the major claims to some extent.

Weaknesses:
My major concern is that it is not clear if the proposed method can serve as a useful method in practice. The paper is not a theory paper and comes with no theoretical analysis, thus I would justify it mainly from the application perspective. My detailed concerns are summarized below:
1. It is not clear if the DLRT algorithm is designed for faster training or faster inference. For faster training, given the evidence in the current draft, I am not convinced that DLRT is even faster than normal training, e.g., DLRT requires conducting extra computation (in lines 8-21 of Algorithm 1, which requires conducting a full SVD per iteration). Not to mention the other baselines, e.g., [1-2], which claim much faster end-to-end running time compared to vanilla training.
2. Memory saving is also claimed in the paper, but I do not believe the low-rank-based method can save memory costs largely during training. Low-rank networks have fewer (or much fewer number of parameters), but the bottleneck of memory cost comes from the layer activations, not model weights [3]. And low-rank neural networks can not reduce the activation size (during training). During inference, low-rank networks can efficiently reduce the memory cost as no activations are required to be saved. The authors should make the claims more clear.
3. The paper does not compare against powerful enough baselines outside the low-rank domain. For instance, quantization+sparsification [5] and learned structured pruning [6] (as low-rank methods are versions of structured pruning after all).
4. The experimental results are too weak, DLRT generates either larger networks than LRNN or with worse accuracy. Then why people should use DLRT in practice given that LRNN method can also adaptively adjust ranks?
5. It has been shown in [2], that the low-rank method won't scale to large tasks, e.g., ImageNet without the proposed warm-up and hybrid network tricks. How does DLRT perform on the ImageNet dataset or even BERT language modeling tasks?
6. It is claimed that DLRT automatically selects ranks, however, it seems the final ranks of the layer are controlled by another hyper-parameter \tau, which is still required to be tuned in practice.

[1] https://arxiv.org/abs/2105.01029
[2] https://arxiv.org/abs/2103.03936
[3] https://arxiv.org/abs/1904.10631
[4] https://ieeexplore.ieee.org/document/9157223
[5] https://arxiv.org/abs/1510.00149
[6] https://openreview.net/forum?id=BJxsrgStvr

---

> ### Author Response · Authors · 2022-08-02
> **Answer to Reviewer kDdv corresponding Paper 7545 (Part 1/2)**
>
> (Part 1/2)
>
> The main concern of the reviewer is that, given that the paper is not theoretical, the paper has to be justified mainly from an application perspective. First, we would like to point out that we have added a whole new section in the new manuscript with two remarkable theoretical results for the proposed method (and a new section in the appendix for their proof), please see the new version of the paper and the response to reviewer j9XY. Thus, we kindly ask the reviewer to re-evaluate our paper in the light of this novel theoretical addition, and not only from the application/experimental points of view.
>
> Concerning your specific concerns, we attempt below to provide a response and clarify why we do not agree with some of them.
>
> 1. This is the comment we are most in disagreement with.
>
> DLRT allows for **both** efficient training and efficient evaluation. However, efficient training is certainly what makes DLRT stand out with respect to available alternatives. To our knowledge, there exists no other paper that provides a method for training directly within the manifold of low-rank matrices, using only the low-rank factors and without ever forming the full operator. Note that while LRNN also adaptively adjusts ranks, it requires the full matrices throughout the training, which are then factorized at each step.
>
> We are sorry this important point was not clear. However, we notice that we have explicitly stated this several times in the paper, including:
>
> - L5-7 (abstract): ``Remarkably, [...] the overall time and memory resources required by both training and evaluation is significantly reduced
> - L41-43 (§1): ``the resulting dynamical low-rank training paradigm [...] requires much less computational cost to be trained''
> - L73 (§2): ``in this work [...] we propose a training algorithm with reduced memory and computational requirements''
> - L108 (§3): ``we propose a method performing the minimization over the manifold of low-rank matrices''
> - etc.
>
> Moreover, we provided several experiments to confirm this. In particular, section 5.1 and figure 4 show exactly that the execution time for both training and evaluation is reduced with respect to the vanilla approach, as long as the ranks are small enough.
> Similarly, Table1 shows that DLRT reduces the number of model parameters to be stored during training up to $96.34\%$.
>
> This is evidence that **Algorithm 1 does not require extra computation** with respect to vanilla training, at least not in general. This can also be predicted with a careful cost analysis, which we conducted in section 4.2 (page 7) and which shows that the overall cost of one forward and one backward step of DLRT is $C_{DLRA}=O(\sum_k r_k^2(n_k+n_{k+1}))$, while the same for a dense network would cost $C_{dense}=O(\sum_k n_kn_{k+1})$. Here $n_k$ are the number of neurons per each layer and we are assuming only one single batch point. In particular, notice that the additional SVD which we have to conduct in each step is made on a $2 r_k\times 2 r_k$ matrix, so for small ranks is a cheap operation. Again, we notice that this has been clearly stated in our (original) manuscript in the *Computational cost* section (line 228).
>
> 2. The point raised by [3] is an interesting one. However, we notice that the memory profiling in [3, Figure~1] does not use any technique to reduce activation memory, which certainly does not represent state of the art treatment of memory intensive applications. We agree that without strategies such as micro batching or checkpointing, the memory bottleneck can be dictated by activation memory. However,  if these techniques are employed (as they are in standard applications), this may not necessarily be the case and relevant  memory requirements do arise from storing the model. Since DLRT reduces both model and optimization costs (in the terminology of [3]) with respect to vanilla training, we disagree with your statement that low-rank training (or sparsity in general) cannot have an impact on  memory costs reduction (when being applied alongside micro batching or gradient taping). We would like to point out that this is also the takeaway of [3] with regard to image classification:
>
> *We find that the appropriate combination of sparsity, low precision, microbatching, and checkpointing
> can lead to a significant reduction in the overall (peak) memory usage of training [...] Incorporating microbatching can boost this to 33.2x with a very minor accuracy loss of 0.09$\%$, and using sparsity can boost this to 60.7x with 0.42$\%$ accuracy loss*.
>
> Moreover, [3] does not discuss settings such as online learning in which DLRT has a direct impact on memory. We do agree however that this point deserves to be addressed in our manuscript to clarify where exactly DLRT tackles the challenges of memory requirements, which is why we have added a sentence in the limitations section.

---

> > ### Author Response · Authors · 2022-08-02
> > **Answer to Reviewer kDdv corresponding Paper 7545 (Part 2/2)**
> >
> > (Part 2/2)
> >
> > 4. We believe this point is highly related to the first one and we strongly disagree. As pointed out on numerous occasions throughout our manuscript, the reduced *training* costs in terms of memory and operations is one of the core property of DLRT.
> > The reason for using DLRT over LRNN is that, unlike LRNN, DLRT never has to compute or store full weight matrices. Training is directly performed on the low-rank factors. Therefore, even if LRNN gives slightly increased accuracy or higher compression rates, it will exhibit costs and memory requirements of the vanilla training method (or worse). Hence, to our understanding from reading the corresponding paper and inspecting the code, LRNN is constructed for efficient inference, while DLRT is constructed for both, inference and learning. We would like to again underline that the singular value decomposition, which has been pointed out by the reviewer to claim inefficiency of DLRT, is performed on a small $2r \times 2r$ matrix, leading to costs of $O(r^3)$. Again, we would like to underline that this has been pointed out in our (original) manuscript in the *Computational cost* section (line 228).
> >
> > 5. See the response to the other reviewers: We did not realize the experimental evidence provided was not enough and we apologize for that. DLRT should perform well if the assumptions made in Theorems 1 and 2 (which we have added in response to Reviewer j9XY) are met. In particular, when the network can be represented efficiently with low-rank weights, we expect our method to find a well-performing low-rank network with the reduced computational costs and memory footprint as discussed. However, we agree that this statement certainly needs to be investigated through further numerical experiments in more complicated settings.
> >
> > Therefore, we already started running several computational experiments on ResNet50, VGG16 and other architectures on several other datasets, including ImageNet, CIFAR100, etc. Given our limited computational resources and the fact that we want to provide error bars for the presented computational results (i.e., we are conducting several runs on every experiment), we don't have all the results yet. We added at the end of the new version (and in the appendix) the preliminary results we have until now, which confirm the findings we observed in the previous experiments. New results will be added as soon as we obtain them.
> >
> >
> > 6. This is a fair point. Note however, that instead of having to pick adequate ranks in every layer (which requires a deep understanding of how the network will be constructed during training) we need to pick a single parameter, which is directly related to the approximation property of the network. If you look at the theorems provided, we can quantify how this parameter affects the overall error, even though this estimate is not sharp and results from a triangular inequality.

---

> > > ### Comment · Reviewer_kDdv · 2022-08-09
> > > **Thank you for the response**
> > >
> > > I would like to thank the authors for providing detailed responses!
> > >
> > > With the theoretical analysis, I think it is safe to increase my overall evaluation score. However, from the practical perspective of view, my major concern still remains, i.e., given [1-2] are already successful, does the proposed method outperform them? If so, can you please add the results to the manuscript? If not, why would people like to use the proposed method?
> > >
> > > I do believe any new method should demonstrate real benefits.

---

> > > > ### Author Response · Authors · 2022-08-09
> > > > **Differences and advantages of DLRT with respect to [1,2]**
> > > >
> > > > Dear Reviewer,
> > > >
> > > >
> > > > Thank you very much for your response. We appreciate your increased score. From your reply, we understand that the main reason for your current evaluation is related to the missing comparison with [1,2]. We believe that significant advantages make DLRT stand out when compared to the mentioned methods. We are again sorry that this was not clear from the main manuscript, and we provide further details below. These new observations have been added to the main paper.
> > > >
> > > > First, our method in contrast to [1,2] allows for **rank-adaptivity**. This is a crucial property that heavily impacts the applicability of low-rank neural networks in general: Having to choose adequate fixed ranks, that is, ranks that yield an optimal compression rate while capturing important structures in each layer, it is a difficult if not impossible task for relevant applications. Having a mechanism that picks these ranks in an automated fashion from a single tolerance parameter, which has a direct explainable and interpretable meaning, is a huge advantage.
> > > >
> > > >
> > > > Second, [1,2] use a straightforward factorization of the form $W=UV^\top$. The proposed updates in DLRT instead are based on the system of differential equation  for the three factors  $K$, $L$ and $S$, which make the training scheme **stable with respect to singular values of the weight matrices**, i.e., allowable learning rates are the same as for the full network. This is supported by our main theorems (Thm1 and Thm2), where the constants in our error estimate and descent rate do not depend on singular values of $S$, which instead would be the case for the  rank factorizations $UV^\top$ used in [1,2].
> > > >
> > > > From the differential-geometry point of view, a major reason why the use of the decomposition in DLRT is more efficient and more stable than the naive  factorization $UV^{\top}$ (as in [1,2]) is that the $K$, $L$ and $S$ steps do not suffer from the high curvature of the low-rank manifold. This is a well-known phenomenon in the differential geometry and time-integration community, see for instance:
> > > >
> > > > [Koch, Lubich, "Dynamical low-rank approximation", Lemma~4.2], and
> > > > [Feppon, Lermusiaux, "A geometric approach to dynamical model order reduction", Section 4 *Curvature of the fixed rank matrix manifold and the differentiability of the SVD truncation*].
> > > >
> > > > As an example of this phenomenon, we have trained Lenet5 on MNIST using DLRT vs the method from [1], fixing all the ranks to 20, except for the last one that is unchanged. In order to highlight the instability of the approach based on $UV^\top$ we used  ill-conditioned starting weights, i.e. we  chose the input weights at random but forcing them to have a very fast singular value decay.
> > > > The table below shows the test accuracy of the two methods over the first 10 epochs. **The method from [1] is heavily affected by the "bad" singular values, while DLRT exhibits robustness and accuracy**, which confirms the statement of our Theorem 1. For both methods we used SGD with fixed 0.01 learning rate and same batch sizes, and we are showing average accuracies over  random splits of the dataset and random ill-conditioned input weights.
> > > >
> > > > | DLRT [%]  | Method from [1] [%]|
> > > > |---|---|
> > > > |53.8667|15.18|
> > > > |80.9632|15.625|
> > > > |86.6594|19.7587|
> > > > |90.5756|20.985|
> > > > |92.8797|21.2124|
> > > > |93.9181|20.8663|
> > > > |94.9862|22.9529|
> > > > |95.1444|27.8283|
> > > > |95.1444|30.7852|
> > > > |96.1531|32.3378|
> > > > |96.2817|35.8881|
> > > >
> > > > Notice that as the same $UV^\top$ decomposition is at the basis of the method in both [1] and [2], we expect an analogous behaviour for [2] (we are currently putting together a test for [2] as well).
> > > >
> > > > To the best of our knowledge, the combination of the two properties of rank adaptivity and robustness with respect to singular values is unique to our work as compared to the available literature. In particular, neither of them is provided by either [1] or [2]. We are convinced that these two properties can heavily impact applications and future research on low-rank neural networks.
> > > >
> > > > Because of all the newly added evidence and theoretical results, we hope that we have clarified your remaining concerns and that you are now convinced of the merit of our work. If this is still not the case, we would be happy to receive further constructive criticism which we will do our best to answer.

---

### Official Review · Reviewer_mpYM · 2022-07-10

**Rating:** 7
**Confidence:** 4
**Soundness:** 3 good
**Presentation:** 4 excellent
**Contribution:** 4 excellent

**Summary:**

This paper presents a dynamic adaptive low-rank training algorithm for deep neural networks that is able to find efficient low-rank subnetworks during training time. It achieves improvement in both memory and computation efficiency. Experimental results on MNIST show that the proposed approach outperforms some other pruning and low-rank methods.

**Questions:**

No other specific questions. Please refer to weaknesses above.

**Limitations:**

Yes. The authors have addressed the limitations and potential negative societal impact.

**Strengths And Weaknesses:**

Strengths
* I really appreciate this work for exploring low-rank training. I think the overall novelty is above the line.
* This paper has detailed equation derivation and algorithms.
* This paper is well organized with sufficient materials to support the idea, especially the sections of computational cost and limitations analysis.

Weaknesses
* Is there a typo at line 298 (LNRR)?
* Why only evaluate on MNIST? How about the performance on more complicated networks and datasets?
* Is there any other related works that research on low-rank manifold and can be compared with in this paper? It would be better if more such comparison is included.

---

> ### Author Response · Authors · 2022-08-02
> **Answer to Reviewer mpYM corresponding Paper 7545**
>
>
> Thank you for carefully reading our work and providing positive and helpful feedback.
>
> 1. Indeed, we had a typo in the name, which has been fixed in the revised manuscript.
>
> 2. We did not realize the experimental evidence provided was not enough and we apologize for that.
>
> Following our discussion on computational costs and memory requirements, we expect DLRT to perform well when the assumptions made in Theorems 1 and 2 (which we have added in response to Reviewer j9XY) are met. In particular, when the network can be represented efficiently with low-rank weights, we expect our method to find a well-performing low-rank network with the reduced computational costs and memory footprint as discussed. We agree with you that this needs to be validated with further  numerical experiments.
>
> Consequently, we already started running several computational experiments on AlexNet, VGG16 and other architectures on several other datasets, including ImageNet, CIFAR100, etc. Given our limited computational resources and the fact that we want to provide error bars for the presented computational results (i.e., we are conducting several runs on every experiment), we
> don't have all the results yet.  We added at the end of the new version (and in the appendix) the preliminary results we have until now, which confirm the findings we observed in the previous experiments. New results will be added as soon as we obtain them.
>
>
> 3. Optimizing within the manifold of low-rank matrices is a recurring theme in Riemannian optimization which has led to numerous remarkable results. Our algorithm is part of this literature but offers a continuous-time point of view.
>
> However, we  are not aware of alternative efficiently implementable methods that perform the training directly on the low-rank manifold with theoretical guarantees of descent and approximation accuracy with respect to the original full-rank problem.
>
> We covered recent low-rank approaches in the numerical experiments section (including the one suggested by Reviewer j9XY). However, the alternatives do not perform low-rank training in the sense they always require the whole weight matrices during training and not only the factors. Thus, they have no memory advantage in the training phase (as highlighted in e.g. Table 1).

---

> > ### Comment · Reviewer_mpYM · 2022-08-09
> > **Thank you for the response**
> >
> > Thank you for the detailed response. My only concern is about the experiments. Thanks for the authors to provide CIFAR experiments in appendix and it seems good to me. Since my rating is already 7, I would like to keep it if no further problems are issued by the other reviewers.
> >
> > Thank you again for your nice work.

---

### Official Review · Reviewer_j9XY · 2022-07-12

**Rating:** 6
**Confidence:** 4
**Soundness:** 3 good
**Presentation:** 3 good
**Contribution:** 3 good

**Summary:**

This paper studies a new training strategy for finding a low-rank sub-network by solving matrix differential equations and tools of matrix decompositions. The idea is novel and the implementation is convincing. However, if this method can be applied to much large scale and frequently used networks instead of current test settings, this work can be definitely more compelling.

**Questions:**

1. Eq 6, what is `min!' ?
2. The description to Fig. 2 is not clear. The timings for what? A gradient descent step, a feed forward pass for a batch of samples?

**Limitations:**

The main limitations are two:
1. Lack theorems or enough theoretical analysis to rigorously support the soundness of the algorithm. Readers may not be familiar with the classical results in related backgrounds. Some very important theory results, like convergence and error analysis, should be clearly stated.

2. I do not understand why the authors do not apply this method in large networks like ResNet50 on ImageNet.

Apart from these two points, I think this is good work. If the authors can successfully address my concerns on these two issues, I will raise my rating.

**Strengths And Weaknesses:**


### Originality
The idea of using matrix differential equations to capture the underlying low-rank weights is novel and attractive. Researchers in many domains have wondered about efficient methods to train the network weights under the factorization of SVDs. So this work may have a broad audience in the machine learning domain. The formulation and implementation of this method also seem to have a rather solid background in numerical analysis and pdes.

### Quality
The basic idea is good, however, the theoretical analysis is a bit limited, and the empirical study is too narrow.
1. The authors should give clear theorems about the convergence and convergence analysis of the proposed algorithm, without which I cannot fully judge the soundness of this work. I am eager to such results in the response of the authors.
2. The authors should also give theorems on what $S_k$ learns during the training, and what is the geometry or intuitive meaning of the final decomposition $USV$ since $S_k$ is not a diagonal matrix,  and $USV$ is not an SVD.
3. Is there any theoretical results that support the universal approximation ability of networks decomposed by this algorithm?
4. Is there any theory analysis on the relationship between hyper-parameters of this method and the test accuracy or approximation error?
5. I am also wondering why the author did not test this method in ResNet, Transformer, or deep generative networks? This seems no barrier to doing so.

### Clarity
The clarity needs many improvements. The authors should summarize the major conclusions in the context into theorems, which can significantly improve the readability and soundness of this work. The background knowledge of this method is also missed, especially the minimum preliminary knowledge for readers to understand the `DLRA' and `KLS'.

### Siginificance
The paper lacks clear theorems to support it theoretical soundness and significance. The numerical results are also a bit narrow for practical usage. There are also some previous methods[3] for leveraging SVD in the training of deep networks, the authors may consider comparing the proposed method with them.

### My suggestions:
1. In Line 105-107, I strongly recommend the authors cite some papers on this topic to enhance the argument. The most relevant two papers could be [1][2]. [1] gives both theoretical and empirical evidence of low rank weight matrix in over-parametric deep networks with tools from recent advances in random matrix theory. Using the theory of dynamic systems and Lyapunov exponents, [2] further describe how chain rules, i.e., large amounts of matrix multiplications can lead to low-rank results and network gradient, which exactly support the motivation here. Besides, [3],[4], and [5] also give support for low-rank networks from different aspects.
2. In line 122-123, I think it would be better to say `$W_k(\tau)\in\mathcal{M}_{r_k}$' for some $\tau>0$. Most networks are initialized with Gaussians, thus initially they have full rank, but there is evidence that training will eventually converge to some low-rank status.

[1]Implicit Self-Regularization in Deep Neural Networks: Evidence from Random Matrix Theory and Implications for Learning. Charles H. Martin Michael W. Mahoney
[2] Rank Diminishing in Deep Neural Networks.  Ruili Feng, Kecheng Zheng, Yukun Huang, Deli Zhao, Michael Jordan, Zheng-Jun Zha
[3] Learning Low-rank Deep Neural Networks via Singular Vector Orthogonality Regularization and Singular Value Sparsification. Huanrui Yang, Minxue Tang, Wei Wen, Feng Yan, Daniel Hu, Ang Li, Hai Li

---

> ### Author Response · Authors · 2022-08-02
> **Answer to Reviewer j9XY corresponding Paper 7545 (Part 1/2)**
>
> Part (1/2)
>
> Thank you very much for the encouraging feedback and for your useful comments and criticism.
>
> We first provide our response to the two main limitation points:
>
> ### Limitations
>
> 1. We did not add much about the theoretical justification of the convergence and accuracy/robustness of the training scheme as it somewhat directly follows from the literature on DLRA and we thought it would have been redundant. However, we agree with your and the other referees' comments this was a bad idea. Also, in writing down the proof details, we realized some passages were less trivial than we originally thought.
>
> We have now prepared a new version of the manuscript where we attempted to complement the lack of theory you have pointed out by discussing some of the main results, together with a proof (in the appendix) that guides the reader throughout the relevant literature and the most careful steps.
>
> For convenience, we also copy the statement of the added theorems below.
>
> =========
>
> Assume the gradient flow $\mathcal F_k(Z) = -\nabla_{W_k}\mathcal L(W_1,\dots,Z,\dots,W_M,\mathcal N(x), y)$ in (2) is locally bounded and locally Lipschitz continuous, with constants $C_1$ and $C_2$, respectively. Then,
>
> **Theorem**
> Fixed $x$ and $y$, let $W_k(t)$ be the (full-rank) continuous-time solution of (2) and let $U_k,S_k,V_k$ be the factors computed with Algorithm 1 after $t$ steps. Assume that the K,L,S steps (7) and (8) are integrated exactly from $0$ to $\eta$. Assume moreover that, for any $Z\in \mathcal M_{r_k}$ sufficiently close to $W_k(t\eta)$, the whole gradient flow $\mathcal F_k(Z)$ is ``$\varepsilon$-close'' to $\mathcal M_{r_k}$. Then,
> $$
> \|U_kS_kV_k^\top - W_k(t\eta)\|_F \leq c_1 \varepsilon + c_2 \eta + c_3 \vartheta / \eta \qquad k=1,\dots, M
> $$
>
> where the constants $c_1$, $c_2$ and $c_3$ depend only on $C_1$ and $C_2$. In particular, the approximation bound does not depend on the singular values of the exact nor the approximate solution.
>
> Observe that, while the loss function  $\mathcal L$ decreases monotonically along any continuous-time solution $W_k(t)$ of (2), it is not obvious that the loss decreases when the integration is done onto the low-rank manifold via Algorithm 1.
> The next result shows that this is indeed the case,  up to terms of the order of the truncation tolerance $\vartheta$. More precisely, we have
>
> **Theorem**
> Let $W_k^t = U_k^t S_k^t(V_k^t)^\top$ be the low rank weight matrix computed at step $t$ of Algorithm 1 and let $\mathcal L(t) = \mathcal L(W_1^t, \dots,W_M^t,\mathcal N(x),y)$. Then, for a small enough time-step $\eta$ we have
>     $$
>     \mathcal L(t+1)\leq \mathcal L(t) - \alpha \eta + \beta \vartheta
>     $$
>
> where $\alpha$ and $\beta$ are positive constants that do not depend on $t$, $\eta$ and $\vartheta$.
>
> ======================
>
>
>
>
> 2. We did not realize our experimental evidence was not enough and we apologize for that. Given the memory and computational cost arguments we have discussed in section 4.2. together with the supporting theory of approximation and descent, we expected a similar behaviour on larger datasets and other NN architectures.
>
>
>
> Nonetheless, our code and the one of a number of baseline alternatives is now running on a range of architectures (including ResNet50, AlexNet and VGG16) and  on several larger and more complex datasets, including ImageNet, tiny-ImageNet, CIFAR10 and CIFAR100. However, given our limited computational resources and the fact that we want to provide error bars for the presented computational results (i.e., we are conducting several runs on every experiment), we don't have all the results yet. We have added a summary of part of the results we have obtained so far at the end of the  new version of the manuscript (with detailed tables reported in the appendix). These results confirm the findings we observed in the previous experiments. We will add new results as soon as we obtain them.

---

> > ### Author Response · Authors · 2022-08-02
> > **Answer to Reviewer j9XY corresponding Paper 7545 (Part 2/2)**
> >
> > (Part 2/2)
> >
> >
> > Second, we provide our response to the questions you raised in the ``quality'' section.
> >
> > ### Quality
> >
> > Concerning the fact that $S$ in $M=USV^\top$ is not a diagonal matrix, this is just another way to parameterize the manifold of rank-$r$ matrices. In fact, one can always consider the svd $S=Q\Sigma P^\top$ of the small $r\times r$ matrix $S$ and retrieve the svd of $M$ as $M=(UQ)\Sigma (VP)^\top$. So, the  subspaces we learn, spanned by $U$ and $V$, are the same as those of the svd of $M$ up to a rotation of the basis (by $Q$ and by $P$, respectively).
> >
> > The geometrical intuition behind the decomposition we learn this way is that during DLRT we are dynamically learning the subspaces $U_k^t$ and $V_k^t$ to be used for compressing/projecting the large network into a lower-dimensional space.
> > The learned factor $S_k^t$ of the decomposition can be interpreted as a tiny low-rank matrix which allows us to extrapolate the main info hidden inside  the model.
> > This can also be seen from Theorem 1.
> > For simplicity,  assume the quantities $\varepsilon$ and $\vartheta$ are small enough and omittable.
> > Because of the invariance of the Frobenius norm under orthonormal transformations, it follows that after $t$ steps of the algorithm we have: $ \| S^t_k - U_k^{t,\top} W_k(t\eta) V_k^t \| = \mathcal{O}(\eta)$.
> > This can be interpreted by saying that the matrix $S^t_k$ (or equivalently its diagonal form $\Sigma_k^t=Q^\top S_k^t P$) catches up to $\mathcal{O}(\eta)$ the information provided by $W_k(t\eta)$ when projected onto the subspace generated by $U_k^t$ and $V_k^t$.
> >
> >
> > Concerning points 3 and 4, these are excellent questions that we are currently investigating. However, their answer is not straightforward in our opinion and we hope we will be able to provide an analysis of the universal approximation and the generalization accuracy in a follow-up work soon. A fist attempt to analyze the influence of the hyper-parameter $\vartheta$ on the approximation error is presented in Theorem 1. There, a preliminary condition on this hyper-parameter can be inferred. A related comment has been inserted in the manuscript.
> >
> >
> > Finally, concerning your **suggestions** and **questions**:
> >
> > - we have taken into account and modified the manuscript accordingly. **However, we could not find details about references [4] and [5].**
> > - we have now added details to the caption of  figure2 (now 1) and rephrased the formula with min!

---

> > > ### Comment · Reviewer_j9XY · 2022-08-03
> > > **Thank you for the response**
> > >
> > > Thank you for the detailed response. I am pleased with the reply. I think this work can be clearly above the bar for acceptance with the newly added theorems and experiment results. I will increase my rating if no further problems are issued by the other reviewers.

---

> > > ### Comment · Reviewer_j9XY · 2022-08-05
> > > **About rating.**
> > >
> > > Dear authors,
> > >
> > > Here is my plan for the rating of this paper. If the newly added experiments can be finished (at least the ResNet on ImageNet) before the discussion period due, and the results seem well, I will increase the rating from 5 to 7. Otherwise, I may only be able to increase it to 6 (weak accept). I also suggest the authors move the cifar experiments to the main context if this paper is accepted.
> > >
> > > Congratulations on your nice work!

---

> > > > ### Author Response · Authors · 2022-08-05
> > > > **Expression of gratitude.**
> > > >
> > > > Dear reviewer, we would like to sincerely thank you for your positive and constructive feedbacks. Your comments have certainly improved the quality of the manuscript. We are currently running the experiments, we hope that our computational resources will be enough to meet the promised deadline. If not, we'll accept gratefully your insightful evaluation.

---

> > > > ### Author Response · Authors · 2022-08-09
> > > > **Updates**
> > > >
> > > > Dear Reviewer,
> > > >
> > > > Again thanks for your report. We took into account your previous comment, and we are going to move the new results to the central part of the manuscript.
> > > >
> > > > We have some partial results for ResNet50 on ImageNet and Tiny-ImageNet. For both datasets, we have trained the network from scratch with same starting weights using DLRT with $\tau = 0.08$ and standard method (we call this "full baseline") with default parameters, no data augmentation, optimizer: SGD with momentum=0.1.
> > > >
> > > > Unfortunately, we are still far from convergence, but after an equal number of epochs we observe positive results:
> > > >
> > > >
> > > >     Imagenet - Resnet50 - train from scratch
> > > >     -----------
> > > >     DLRT:
> > > >     epochs: 21
> > > >     top1 test acc: 51%
> > > >     top5 test acc: 76.7%
> > > >     training compression rate: 55.4%
> > > >
> > > >     full baseline:
> > > >     epochs: 21
> > > >     top1 test acc: 51.4%
> > > >     top5 test acc: 77.01%
> > > >
> > > >
> > > >     Tiny-imagenet - Resnet50 - train from scratch
> > > >     -----------
> > > >     epochs: 90
> > > >     top1 test acc: 54.9%
> > > >     top5 test acc: 80.2%
> > > >     training compression rate: 55.2%
> > > >
> > > >     full baseline:
> > > >     epochs: 90
> > > >     top1 test acc: 55.2%
> > > >     top5 test acc: 78.56%
> > > >
> > > > We will continue running the experiments in the next days and include additional tests in the main context (together with the results on cifar) if the paper will get accepted.
> > > >
> > > > We would also like to point your attention to the additional test on stability we performed in response to the comments of reviewer kDdv, and the overall effort to improve the theoretical and experimental evaluation.
> > > >
> > > > We sincerely hope this additional effort will convince you of the merit of our work!

---

### Meta-Review · Area_Chair_H7j1 · 2022-08-25

**Recommendation:** Accept
**Confidence:** Certain

**Metareview:**

This paper proposes a Dynamic Low Rank Training Scheme (DLRT) to optimize the neural network weight matrix imposing low rank constraint and hence yields better computational and memory efficiency. The obtained solution becomes low rank and thus it can realize factorization of the weights by its nature. The optimization procedure can be equipped with an adaptive rank selection scheme. The proposed method is mainly justified by numerical experiments.

The optimization method is basically derived from the gradient flow along the low rank matrix manifold. This paper gives a rather theoretically solid optimization scheme.
The presentation is overall good. It is well organized and contents required to understand the contribution are appropriately presented.

There are still some weakness. First, the paper would benefit from adding some more related topics and giving more discussions about connection to them. Second, it would enhance the paper if the additional numerical experiments on larger data and models (Imagenet/Resnet50) as presented in the rebuttal phase would be included.

In summary, although there are some weakness, this paper gives a novel and solid methodology to obtain a low rank weight matrices. So, I recommend acceptance.
On the other hand, I strongly recommend the authors to address the issues pointed out by the reviewers in the final version.


**Award:**

No

---

### Decision · Program_Chairs · 2022-09-14

Accept